# Arrangement and symmetry of the fungal E3BP-containing core of the pyruvate dehydrogenase complex

B. O. Forsberg [1], S. Aibara[1,2], R. J. Howard [1], N. Mortezaei[1,3] & E. Lindahl[1,4 ✉]

The pyruvate dehydrogenase complex (PDC) is a multienzyme complex central to aerobic respiration, connecting glycolysis to mitochondrial oxidation of pyruvate. Similar to the E3-binding protein (E3BP) of mammalian PDC, PX selectively recruits E3 to the fungal PDC, but its divergent sequence suggests a distinct structural mechanism. Here, we report reconstructions of PDC from the filamentous fungus *Neurospora crassa* by cryo-electron microscopy, where we find protein X (PX) interior to the PDC core as opposed to substituting E2 core subunits as in mammals. Steric occlusion limits PX binding, resulting in predominantly tetrahedral symmetry, explaining previous observations in *Saccharomyces cerevisiae*. The PX-binding site is conserved in (and specific to) fungi, and complements possible C-terminal binding motifs in PX that are absent in mammalian E3BP. Consideration of multiple symmetries thus reveals a differential structural basis for E3BP-like function in fungal PDC.

[1] Department of Biochemistry and Biophysics, Science for Life Laboratory, Stockholm University, 17165 Solna, Sweden. [2] Department of Molecular Biology, Max Planck Institute for Biophysical Chemistry, 37077 Göttingen, Germany. [3] Vironova AB, 11330 Stockholm, Sweden. [4] Department of Applied Physics, Swedish eScience Research Center, KTH Royal Institute of Technology, 17168 Solna, Sweden. ✉email: erik.lindahl@dbb.su.se

Anaerobic glycolysis catabolizes glucose through a series of electron transfer reactions that produce adenosine triphosphate and the terminal electron acceptor pyruvate. The pyruvate dehydrogenase complex (PDC) catalyzes the further formation of acetyl-coenzyme A (acetyl-CoA) from pyruvate, enabling the tricarboxylic acid cycle that sustains oxidative phosphorylation. Consequently, deficiency or malfunction of the PDC profoundly impacts metabolic fitness and normal development[1], and is also associated with severe metabolic disorders, such as Leigh syndrome and episodic ataxia[2]. In cancer biology, the PDC attracts attention because aerobic fermentation is upregulated in many cancers, while inhibition of the PDC results in decreased mitochondrial glucose oxidation, known as the Warburg effect[3,4]. Decreased mitochondrial activity and reduction of reactive oxygen species generated by the PDC[5] both inhibit fundamental apoptotic pathways[6], whereas activation of the PDC in symptomatic melanomas has in fact been shown to restore mitochondrial activity and induced senescence[7].

The PDC is a multienzyme complex consisting of three components: E1, E2, and E3. E1 catalyzes the decarboxylation of pyruvate and transfers the resultant acetyl group to an E2 carrier domain. E2 then covalently links this acetyl group to CoA through its catalytic domain. Finally, E3 catalyzes the reoxidation of the E2 carrier domain lipoyl moiety, allowing the chain of reactions to repeat[8]. Without exception, the PDC is organized by non-covalent recruitment of E1 and E3 to a large core built from E2 catalytic domains. The co-localization of the participating enzymes and substrates increases the overall rate by minimizing substrate diffusion, and tightly couples the chain of reactions[9]. This in turn enables rapid and reversible regulation of the reaction cycle through (de)phosphorylation of E1 by phosphatases and kinases that are weakly bound to the PDC[9], adapting its activity to current metabolic requirements[10,11].

The tethering of E1 and E3 to the E2 core by linking regions that are likely flexible and arranged less symmetric than the core makes it challenging to reconstruct faithfully[12,13]. The only current viable option is single-particle cryo-electron microscopy (cryo-EM), but even then it remains challenging, since tethered components manifest mainly through the probability distribution of their position around the core. This obscures any structural characteristic, so that any such reconstruction is not possible to interpret as faithful low-pass filtered reconstructions, as previously assumed[12–14]. Due to these factors, a global view of fully assembled PDC, or comprehensive knowledge of its flexibility and transient contacts has never been presented for any species. Nevertheless, structures of isolated PDC components[9,15], and core assemblies with high symmetry[16–18], have been determined.

The saturation and relative proportions of tethered E1 and E3 may also be influenced by availability of tethered or tethering components, which in turn may change in response to expression levels and alteration in metabolic requirements. In mammals, this is further exploited through the expression of a catalytically inactive E2 variant with viable substrate-carrier domains[19] called E3-binding protein (E3BP) that partially substitutes E2 and selectively recruits E3 (refs. [20,21]). The fungal PDC also expresses such an E2-homologous protein to recruit E3 (refs. [22,23]), originally known as protein X or the X-component (PX). There is however no obvious similarity in the C-terminal domains between mammalian E3BP and fungal PX. This suggests that a reduction in E2 activity by replacement with catalytically inactive subunits is unlikely in fungal PDC, and decoupled from E3 recruitment. It has not yet been possible to show how these E3-specific components assemble in the PDC core for any mammalian or fungal species, but models and stoichiometric measurements have suggested E2:PX stoichiometries of 2:1 (ref. [24]), 4:1 (refs. [25,26]), and 5:1 (ref. [27]). The challenge to determine how

this regulation is dictated by structure lies in the suppression of relatively small asymmetric features by a largely symmetric complex, here termed implicit symmetrization. This is a topic of ongoing research, largely focused on coinciding, but incompatible symmetries[28], termed symmetry mismatches.

To show how fungal PX is incorporated into the core assembly and how its stoichiometry is regulated, we present a structural analysis of the PDC from the fungal model organism *Neurospora crassa*. We consider the implicit symmetrization caused by the icosahedral E2 core and are able to determine tetrahedral reconstructions, showing how the fungal PDC departs from the dominating symmetry without introducing symmetry mismatches. The tetrahedral-symmetric arrangement of PX interior to the icosahedral PDC core dictates a complex composition similar to that suggested for mammalian PDC. Based on its oligomeric state and size, we also identify factors that offer a simple explanation for the range of previously presented PDC component ratios in fungi. We confirm that PX is critical to *N. crassa* pyruvate metabolism, and that by extension the stoichiometric ratio of the complex components is pivotal to overall activity. Any change in PX oligomer stability and/or size would therefore affect PDC component stoichiometry, which could be utilized to allow fungi to tune PDC component activity in a dynamic fashion. Further, we discuss the fidelity of the established classification and exemplify how marginalized reconstruction leads to uncertainty in the classified proportions of cryo-EM particles, which is crucial to the interpretation of the present results.

## Results

**Structure determination of a non-icosahedral *N. crassa* PDC core component.** In order to characterize the fungal PDC, we purified intact complexes from *N. crassa* mitochondria and reconstructed its native structure from single-particle cryo-EM data. This showed a dodecahedron-shaped core with icosahedral symmetry, composed of 60 E2 core-forming domains decorated with many flexibly tethered enzymes surrounding it (Fig. 1a). To improve the reconstruction, icosahedral symmetry was enforced. This reconstruction allowed identification of E2 residues 232–458, including part of the N-terminal linker that extends toward the recruitment domain (peripheral subunit-binding domain, PSBD). Interior to the core assembly, we found an apparently ordered, non-icosahedral PDC component (Fig. 1a), which appears consistent with a mixture of states, incorrectly averaged under icosahedral symmetry. In order to reconstruct this component faithfully, we attempted to avoid over-symmetrization through use of a lower symmetry group. All icosahedral symmetries were attempted. While this alone did not aid the interpretation of the interior density for any such sub-symmetry, 3D classification coupled with the application of tetrahedral symmetry yielded a correctly averaged reconstruction symmetry, evidenced by a protein-like density (Fig. 1d) and consistent background in the complementary interior volume. A preliminary analysis of the interior assembly suggested that an alternate tetrahedral configuration should exist with equal probability. Through repeated classification using only the core assembly as an initial reference, such a state was indeed observed. Three distinct classes were consistently observed under tetrahedral symmetry, partially reconciling the over-symmetrized appearance of the internal density under icosahedral symmetry (Fig. 1b). All such classes show an identical E2 core scaffold with the same handedness, differing only in their interior content. The first class has four interior, threefold symmetric densities, each immediately interior to an E2 trimer. Each such threefold interior density appears as a basket that hangs underneath (interior to) an associated E2 trimer. They are arranged such that they associate

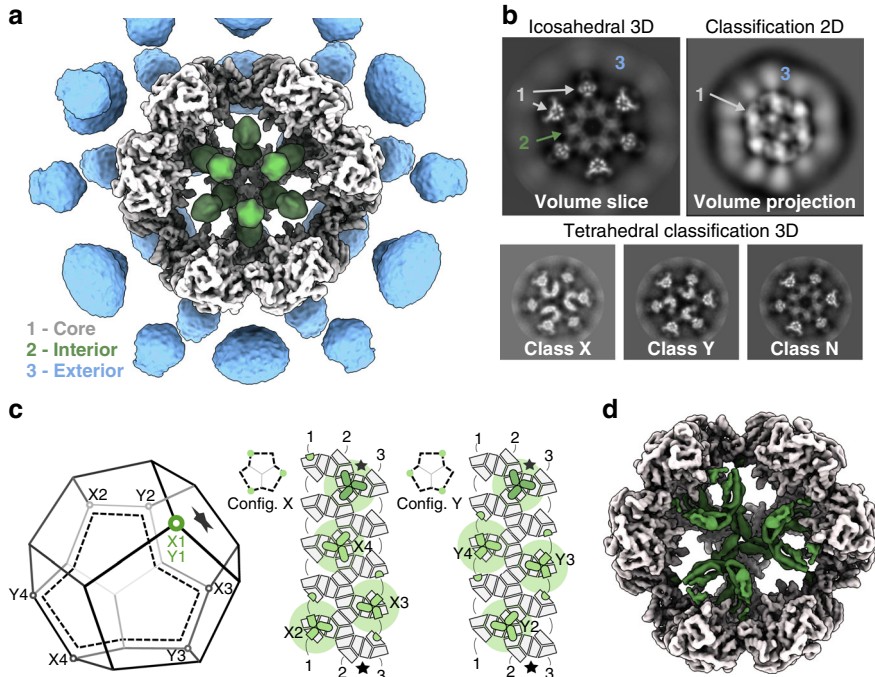

**Fig. 1 The _N. crassa_ PDC has a core-internal protein with tetrahedral optimal packing. a** Reconstruction of the intact native PDC imposing icosahedral symmetry shows symmetrization artifacts, both exterior (blue) and interior (green) to the core (gray), predominantly along fivefold (interior and exterior) and twofold (exterior) symmetry axes. **b** Whereas the exterior artifact appears to represent a continuous degree of freedom resulting in a blurred but elevated density, the interior component is consistent with a discrete mixture of states. Classification of the PDC as tetrahedral resolves this mixture into three classes of interest: tetrahedral classes X and Y, and class N that still appears to be over-symmetrized. **c** A schematic dodecahedron (left) illustrates which four E2 trimers (vertices) are occupied by interior density in each of classes X and Y. These arrangements are also shown mapped to an unfolded view of the E2 core (right), where each E2 monomer is shown in gray, and each interior monomer is shown in green. These monomers form four threefold-symmetric oligomers with equivalent positioning (also Supplementary Fig. 1). The effective volumetric occlusion of the core interior is schematically illustrated by a background circle—two interior components cannot be placed closer than shown here. **d** A cutaway view illustrates the tetrahedral class Y reconstruction, and the protein-like density (green) interior to the PDC core.

with E2 trimers along one direction of each threefold axis of the tetrahedral symmetry (Fig. 1c). The second class shows interior densities identical to the first, but arranged with E2 trimers along the opposite direction of each threefold symmetry axis, resulting in a distinct relative arrangement that cannot be equated to the first through any rigid body transformation. These classes are therefore denoted tetrahedral classes X and Y, respectively. In neither of these classes does the trimeric assemblies appear to come into physical contact. The third class appears to be over-symmetrized under tetrahedral symmetry (Fig. 1), similar to that seen under icosahedral symmetry. In this paper, we term this the non-tetrahedral class N. To approach a quantification of the distribution of data across these classes, a three-class classification was conducted using the above reconstructions as simultaneous and separate input references. This assigned ~30% of the input particles to either of the classes X and Y, and the remaining 40% to class N.

To explain the prevalence of the tetrahedral arrangement in native PDC, we next evaluated what steric restrictions are associated with the interior density volume occlusion. We computationally isolated a single threefold interior density, and examined that configurations of multiple such densities result in clashes. We find that no two E2 trimers separated by fewer than two intermediate-E2 trimers within the core scaffold can accommodate a structured interior basket density (Supplementary Fig. 1d). Under this restriction it is possible to arrange four such interior densities in two unique arrangements (Fig. 1c, config. X and Y). These coincide with tetrahedral symmetry and correspond to each of the classes X and Y found by applying

tetrahedral symmetry during cryo-EM classification. As such, we find that interior assemblies do not need to make contact to arrange as observed, and therefore postulate that they do not. Note that we use the nomenclature class to describe a data subset, and configuration or arrangement for the idealized spatial organization of the interior density.

To further examine class N, that represents a seemingly non-tetrahedral class, we considered how the tetrahedral symmetry of the interior might break. First, it is possible to arrange three or even two interior trimers such that further additions can be made without incurring clashes (Supplementary Fig. 1e, config. 3S and 2S). We term these configurations suboptimal. Second, if one or more available locations are unoccupied the interior would be unsaturated (e.g., Supplementary Fig. 1e, config. Y3U). Third, dimeric or monomeric interior densities are possible. Finally, a combination of any of the above cannot be ruled out. Attempts to reclassify the 40% of particles assigned to the non-tetrahedral class N under any tetrahedral sub-symmetry did not yield consistent results, but occasionally resulted in one or both of the tetrahedral classes as an implicitly reinforced sup-symmetry, when permitted (D2, C3, C2, and C1). In other symmetries (D5, D3, and C5), the core is faithfully represented but the interior component appears over-symmetrized. We took this to indicate that class N contains particles with an unsaturated, but optimally arranged interior, that tends to dominate reclassification and manifest as tetrahedral-like through explicit and/or implicit symmetrization. Any additional conflicting particles may be hidden by broad backprojection (see "Discussion" section). No class of particles displaying suboptimal saturation or asymmetric

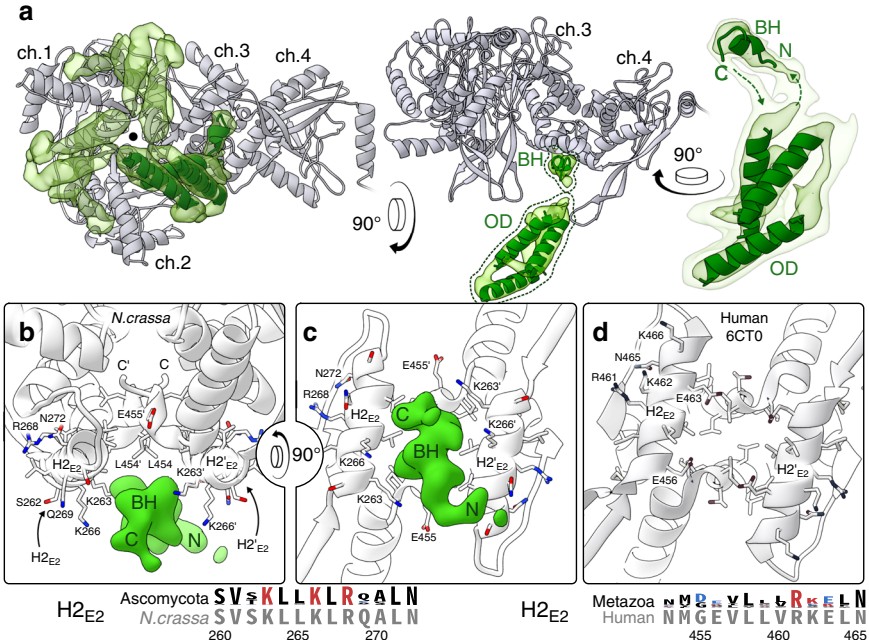

**Fig. 2 The E2-pair interface constitutes a conserved basic/hydrophobic binding pocket. a** E2 chains (gray) 1–3 form a threefold-symmetric oligomer, and each of them additionally forms a twofold-symmetric bridge to an adjacent E2 monomer (here labeled chain 4). A pocket inside each of these three twofold E2 bridges accommodates one binding helix (BH) of the threefold interior density (transparent green). The bulk of the interior density forms the oligomer domain (OD) that extends N-terminally from the BH. The OD is consistent with a three-helix bundle, although secondary structure and connectivity remain ambiguous. Helices of ~20 residues are docked to illustrate this (green cartoon). **b** The BH binds asymmetrically to the symmetric pocket formed by the E2 bridge (Supplementary Fig. 3). **c** A rotated view shows that in Ascomycota, pocket-lining lysines in H2$_{E2}$ are conserved (sequence logo). **d** In contrast, the human E2 interface is less basic and conserved, apart from residues involved in substrate interaction (PDB:6CT0).

baskets was confidently identified, perhaps because the complete asymmetry of any such configuration would be outcompeted by implicit symmetrization in the presence of the E2 core. Taken together, these observations indicate that the interior component prefers a threefold-symmetric state, arranged according to an optimal geometry and near saturation under the current (native) conditions.

We next refined the tetrahedral classes X and Y individually. The resolution of the interior density varies between 4.0–6.5 Å as assessed by local Fourier shell correlation (Supplementary Fig. 2). Following local resolution filtering, we confirm that classes X and Y both depict the same threefold-symmetric oligomer interior to the core. The only observable points of contact with the E2 core are in the twofold interface bridges connecting the density-associated E2 trimer to each of its three neighbors (Fig. 2a). E2 here provides a predominantly hydrophobic pocket, composed of the C-terminal residues that form the bridge and helix 2 (H2$_{E2}$; Fig. 2d). Notably, while H2$_{E2}$ does not appear to be conserved among metazoa, its conservation in Ascomycota indicates an essential function (Fig. 2d, e). Here, H2$_{E2}$ contains a KLLK motif (K263–K266) with basic lysine residues oriented toward the interior density, although precise contacts remain to be identified. Additional conserved residues, including R268 and N272, interact with the substrate CoA[29] rather than contributing to the interior-density interface, and should therefore not be considered constituents of the binding pocket (Fig. 2b, c).

The segment of the interior density that binds to the E2 bridge pocket comprises a small (approximately seven residues) alpha-helical segment, here referred to as the binding helix (BH). The backbone and direction of the BH could be traced from our reconstructions (Fig. 2b), but the resolution did not permit assignment of its primary sequence. The BH connects via its N-terminal end to the body of the threefold interior oligomer domain (OD), and via a weaker density to its C-terminus

(Fig. 2a). Owing to its low reconstructed resolution, the precise fold of the OD is unclear, but it appears to be most consistent with a bundle of three alpha-helices of ~20 residues each (Fig. 2). The relatively low resolution of the OD density suggests it has some flexibility with respect to the E2 core. While this flexibility is expected given the delicate connection to the BH, the oligomer evidently supplies a sufficiently rigid or persistent steric obstacle to impose a tetrahedral symmetry on a large portion of the native PDC particles. Its small size, the high noise contributed by the flexible periphery, and the difficulties associated with symmetry unfortunately preclude a complete analysis its flexibility, and this restraint using our data and reconstruction.

**The core-interior density is a partial C-terminal domain of PX.** We next sought to confidently identify the protein occupying the PDC core interior. Mass spectrometry (MS) detected E1α, E1β, E2, E3, and PX, as well as kinases and phosphatases known to regulate the PDC (Supplementary Fig. 4). E1 and E3 were excluded from consideration based on homologous structures being known and incompatible. To exclude both the lipoyl domain (LD) and PSBD of E2, we recombinantly expressed and purified *N. crassa* E2, and obtained a single-particle cryo-EM reconstruction. In contrast to the endogenous PDC, the interior of the recombinant E2 assemblies showed no evidence of a structured density subject to over-symmetrization (see Fig. 3a, 5) under icosahedral symmetry. The interior did display a higher overall density compared to pure solvent, consistent with a flexible component, similar to the PDC exterior. We attribute this to the N-terminal PSBD and LD being free to diffuse into the core interior.

It seemed plausible that the C-terminal domain of fungal PX, which is homologous within fungi but not to the metazoan E3BP C-terminal domain, could account for the interior density. We

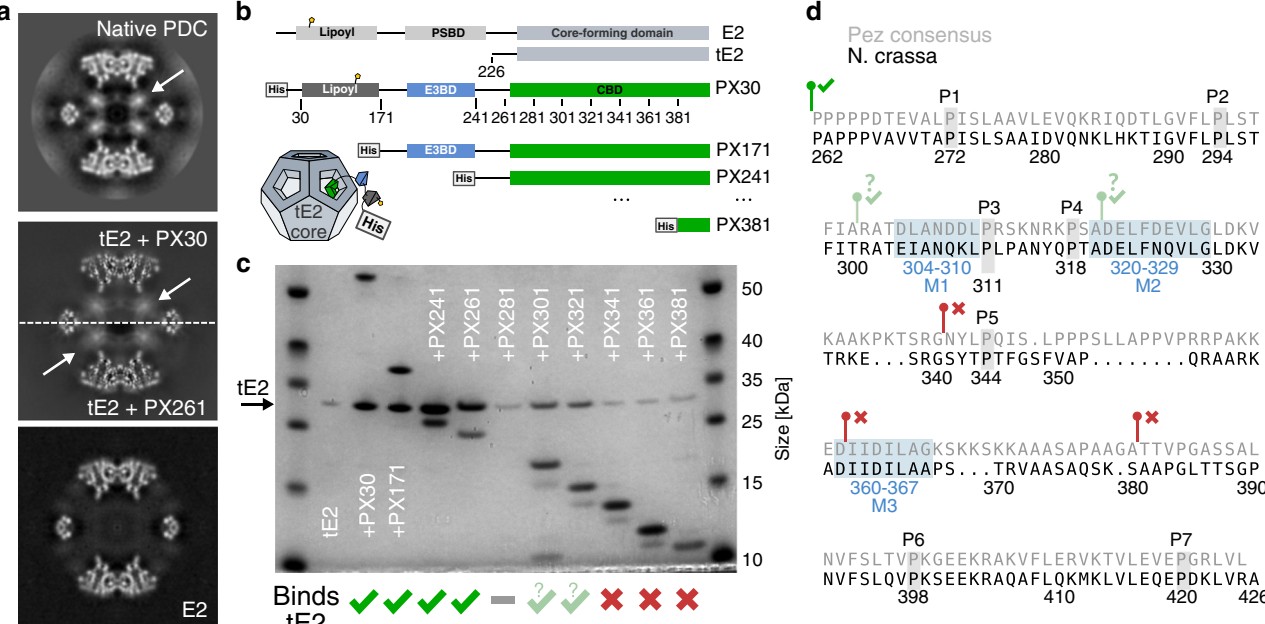

**Fig. 3 N. crassa PX is a monomer that oligomerizes upon binding to the PDC core. a** Comparison of over-symmetrized interior densities. The native PDC (top) was the only dataset wherein a tetrahedral arrangement of interior densities could be classified confidently. It is however clear that both a reconstituted tE2 + PX30, as well as a co-expressed tE2 + PX261 subcomplex (middle) display similar interior density signatures under icosahedral (over-) symmetrization. The E2 core alone (bottom) does not show this signature. **b** Schematic of recombinantly co-expressed tE2 and PX constructs used for binding assays. PX<N> indicates truncation of residues 1–N. **c** SDS–PAGE showing co-purification of E2 core assemblies by His-tag affinity of PX constructs, illustrating a disproportionate decrease in bound E2 upon truncation past residue 280. PX281 exhibited a complete loss of expression or rapid degradation. Further truncation restored PX expression, but at reduced affinity or increased ratio. Binding was completely abolished (comparable to lone tE2, lane 2) following N-terminal truncation past residue 340. **d** PX CBD sequence from N. crassa aligned to the the Pezizomycotina (Pez) consensus sequence. The seven conserved prolines are indicated, as well as the three conserved $H2_{E2}$-complementary motifs M1–3, found in Pez. Notably for N. crassa, M1 does not agree well with the consensus motif (cf. Supplementary Fig. 7). Green and red markers indicate the starting positions and binding of PX constructs corresponding to **c**.

therefore carried out separate recombinant expression of N. crassa E2 core-forming domain (tE2) and His-tagged PX variants, which could be co-purified by Ni-NTA affinity following brief incubation. Neither the full-length PX nor any N-terminal truncation construct of PX appear to form stable oligomers in solution and in absence of the core-forming E2. The C-terminal domain of PX however appears to bind to the icosahedral assembly of fungal E2 (Fig. 3). To confirm that this binding localizes to the E2 interior, we collected single-particle cryo-EM data of a reconstituted tE2 + PX recombinant subcomplex. The reconstruction shows an interior density that under icosahedral symmetry (i.e., when over-symmetrized) appears identical to that observed in the endogenous preparations (Fig. 3a and Supplementary Fig. 6). The overall resolution is better, likely due to the absence of the PDC exterior. However, we are not able to find a tetrahedral class of particles in this or any recombinant preparation, which we attribute to a decreased occupancy and order of the interior assembly compared to the endogenous preparation. The increased resolution of the E2 region may also dominate alignment and increase implicit symmetrization, further de-emphasizing the interior component and obscure the sub-symmetric interior. The appearance of the over-symmetrized interior nonetheless unambiguously shows that the interior density observed in the native PDC reconstruction can be accounted for by the C-terminal domain of PX, namely the core binding domain (CBD). We also attempted to subtract the core and classify only the interior PX density. However, the periphery and low order of the endogenous and recombinant preparations, respectively, prohibited this approach from supplying any additional information regarding the fungal PDC interior.

To identify the possible regions of the CBD responsible for interaction with the E2 bridge, we co-expressed His-tagged N-terminal PX truncations alongside an untagged E2 core-forming domain, inspired by a prior investigation[30]. Affinity purification on Ni-NTA resin indicated co-purification of E2 by PX constructs omitting LD and PSBD, but including the full CBD (Fig. 3c, lanes 3–6). N-terminal truncations of PX beyond D280 appear to alter the interaction, despite equivalent expression of the PX construct (Fig. 3c, lanes 7–12). This indicates that the CBD is required in its entirety to maintain its functional fold within the E2 core assembly, possibly affecting oligomerization and/or binding. Specifically, constructs PX301 and PX321 pulled down diminished levels of tE2. We attribute this to either decreased affinity for E2 or increased PX:E2 stoichiometry. Disruption of oligomerization may thus have caused an increase in bound PX by omission of the steric condition imposed by core-interior oligomerization. Truncation past G340 however abolished binding completely (cf. Fig. 3c, lanes 2, 10–12). This suggests that a critical binding motif is localized N-terminal to G340.

**Domain architecture of the CBD.** To further map the CBD, we identified 341 homologs of PX from various fungal species. Most share the expected domain topology LD–PSBD–CBD, while some Eurotiomycetes (Eur) sequences lack the LD (Supplementary Fig. 7). Based on sequence conservation, the PSBD seems to extend C-terminally compared to, e.g., human E3BD, leaving only a short linker evidenced by low sequence conservation (Supplementary Fig. 7). This might suggest increased order of E3 in fungi, but we observe no quantifiable order of the periphery to corroborate this. Moreover, the limited sequence similarity of

fungal and mammalian PX/E3BP (Supplementary Fig. 8), led us to analyse the fungal CBD without consideration for mammalian E3BP sequence data. The sequence similarity in the fungal CBD indicated a suitable clustering of Pezizomycotina (Pez) separate from Saccharomycotina (Sac), and a small set of dissimilar sequences. The latter set was omitted, and separate alignments were made for Pez (including *N. crassa*, 260 sequences) and Sac (including *Saccharomyces cerevisiae*, 61 sequences). Alignment of Pez sequences delineate the its CBD by seven conserved prolines P1–P7 (Fig. 3d and Supplementary Fig. 7), and a proline-rich region of lower conservation between P5 and P6. Here, a stretch of eight conserved residues form a DJJDϕLxG-motif (J = I/L, ϕ = hydrophobic, x = any) with the consensus sequence DIFDLLAG (Fig. 3d, motif M3). This motif was predicted as possibly helical, and could correspond to an acidic BH complementary to the basic/hydrophobic $H2_{E2}$ pocket (Fig. 2). Notably, alignment of Sac sequences revealed the same motif. Alternatively, two similar conservation motifs are found in Pez, just before P3 and just after P4 (Fig. 3c and Supplementary Fig. 7c, motifs M1–2), although their charge is less conserved. N-terminal truncation of the second of these motifs coincides with abolished binding (Fig. 3c). Definitive assignment of the BH will however require more detailed biochemical and/or structural investigation. Multiple additional conservation patterns however also exist, which unfortunately require a better structural assignment of the PX primary sequence to be confidently interpreted.

One long helix was predicted between P6 and P7 (Supplementary Fig. 7c), likely comprising part of the OD observed by cryo-EM. Interestingly, beta strands were predicted and corroborated by a strong coevolutionary signal, indicating a two-strand sheet (Supplementary Fig. 7). While there was no obvious correspondence between the reconstructed OD and the predicted secondary structure elements, a coevolutionary analysis of the Sac-alignment indicated a similar and partially coinciding three-strand sheet. Together with the conservation of the M3 motif (DJJDϕLxG) in sequences of otherwise low similarity, our observations indicate that (i) the M3 motif is a likely candidate for the BH that is shared across Ascomycota, and (ii) the overall fold of the OD and/or entire CBD may differ in *N. crassa* vs. *S. cerevisiae*. It is also possible that multiple motifs from each PX monomer occupy multiple binding sites to limit binding stoichiometry; however, we see no conclusive indication of this in our cryo-EM reconstructions.

**N. crassa PX is essential for PDC function.** To confirm the impact of PX on glucose metabolism, we assayed activity of PDC purified from wildtype and PX-knockout *N.crassa*. We first confirmed that wild-type PDC requires pyruvate, NAD, and CoA for activity, as well as supplemented cofactors thiamine pyrophosphate (TPP) and $Mg^{2+}$. Enzymatic activity to 0.5 μmol NADH $min^{-1}$ $mg^{-1}$ was two orders of magnitude higher than reported for reactivated mammalian tissue extracts[31], but an order of magnitude lower than what has been reported for PDC isolated from cauliflower[32] or *Zymomonas mobilis*[33]. These discrepancies may be attributed to differences in PDC inactivation or assay temperature (see "Methods" section), but may also stem from differences in overall complex organization or composition. Conversely, PDC purified from the PX-knockout showed a complete loss of activity (Supplementary Fig. 4). MS confirmed the absence of PX and E3, suggesting that the loss of PDC activity is due to disrupted E3 recruitment. In a differential growth assay, knockout growth on 1% Na-acetate was similar to wildtype, but nearly ablated on 1% sucrose (Supplementary Fig. 9). Inhibited growth on sucrose but not Na-acetate indicates a metabolic dysfunction attributable to impaired PDC function. This indicates

that the PX-knockout relies on free mitochondrial E3, which drastically impairs pyruvate metabolism through the PDC in vivo. Future studies are however necessary to investigate to what extent free E3 can rescue function. Wildtype and knockout strains showed comparable E1α-phosphorylation of S317, which has been shown to deactivate the human PDC[9,34] (Human-S293, or S264 in some references). This indicates that suppressed growth did not arise from differences in kinase/phosphatase recruitment. A more detailed MS analysis may be necessary to corroborate this, and elaborate on possible alterations caused by enzyme modification or degradation.

**PX arrangements are incompletely classified due to marginalization.** Strictly enforcing a symmetry makes any sub-symmetry over-symmetrized. Sup(er)-symmetric reconstructions can occur, wherein a higher symmetry than specified is reconstructed. This is most obviously the case when a partial sup-symmetry is present, e.g., the icosahedral core around the tetrahedral PX interior. This effect is enhanced when multiple orientations are utilized in a weighted manner for the backprojection of each particle, known as marginalization of the alignment. This leads to implicit symmetrization, which may obscure lower symmetries in favor of sup-symmetries. For this reason, specifying icosahedral sub-symmetries is not a rigorous method to classify PDC particles according to interior arrangement, despite its success in identifying two tetrahedral arrangements. In fact, no good method exists to classify particles according to symmetry.

The portion of the fungal PDC particles that are not identified as tetrahedral presumably have any of a number of possible interior arrangements with lower degrees of symmetry (Supplementary Fig. 1). Due to the described complication of symmetry-guided classification, we are unable to reconstruct any interior arrangement other than the tetrahedral one. We do however consistently observe a tetrahedral sup-symmetric reconstruction using data that was not originally identified as tetrahedral. This indicates that marginalization affects classification in a symmetry-dependent manner. The larger alignment space of lower symmetries correlates with the capacity to hide data by marginalization, which would affect classification results. This leads us to speculate that the sup-tetrahedral class is in fact not built from tetrahedral particles, but from particles that under a lower symmetry constitute an identifiable near-tetrahedral (or otherwise compatible) subset due to the higher capacity of marginalization to hide conflicting data. This would indicate that at least part of the non-tetrahedral particles have lost one or more PX trimers, as opposed to having PX trimers in an unfavorable (non-tetrahedral) configuration (cf. arrangements S2 and S3, Supplementary Fig. 1).

Furthermore, we cannot hope to quantify the relative proportions of interior arrangements even if these were identifiable. To illustrate how marginalization and implicit symmetrization prevent this, we recombined data pertaining to the distinct tetrahedral arrangements X and Y, in equal proportion, and ran a tetrahedral 3D refinement with a mask made from the union of masks covering either tetrahedral arrangement. This reflects incompletely classified data with similar but irreconcilable subpopulations. We find that one of the arrangements overpowers the other, as the reconstruction is assuredly not the superposition of both arrangements (Supplementary Fig. 10). There is thus no indication of heterogeneity, which may lead to false inferences regarding the homogeneity of the classified subset. The superposition of the classes that clearly indicate heterogeneity can be recovered, but only when marginalization is entirely eliminated and alignments established for each class individually are used. This is thus a strong indication that marginalized reconstruction

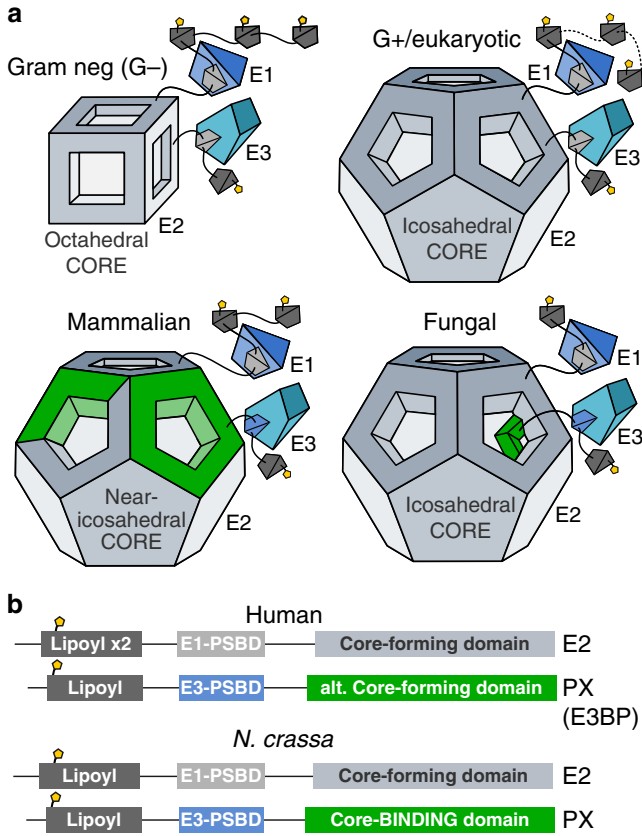

**Fig. 4 The fungal PDC is analogous in function to the mammalian PDC, but structurally distinct. a** Previously known PDC assemblies are octahedral and icosahedral core assemblies surrounded by flexibly tethered enzymes. The mammalian PDC has been shown to include an alternate core-forming protein that specifically recruits E3, but where an unknown mechanism and/or arrangement supplies a stoichiometric regulation to 40:20 or 48:12. The fungal PDC appears to also recruit E3 separately, but through a mechanism that maintains a homomeric core with strict icosahedral symmetry. The additional protein in the case of *N. crassa* forms a tetrahedral assembly interior to the core under optimal conditions, which results in a binding stoichiometry in direct correspondence to the icosahedral and tetrahedral symmetries, i.e., 60:12. The fungal PDC thus appears to have arrived at a similar stoichiometry and functional equivalence as the mammalian PDC, through an entirely distinct structural mechanism. **b** The domain topology and function of mammalian and fungal PX are strikingly similar, despite their disparate structural rationales.

risks hiding large portions of data, and that it is consequently impossible to directly interpret any class population that has not been validated by other methods.

## Discussion

The cryo-EM reconstruction of the *N. crassa* PDC core exhibits strict icosahedral symmetry, but with an additional protein PX that forms a tetrahedral assembly interior to the core under optimal conditions. Two distinct tetrahedral configurations could be identified and refined to ~4 Å resolution. Despite a completely different structural mechanism, we confirm that the fungal PX component mirrors the mammalian E3BP as a selective recruiter of E3. Rather than substituting core E2 domains, PX oligomerizes interior to the PDC core (Fig. 4), occupying a significant portion of the available space. The oligomerization and volume occlusion by PX naturally suggest a mechanism for the variation of binding stoichiometry previously found for *S. cerevisiae*. We show conclusively that throughout Ascomycota, PX binding stoichiometry

is limited by the 30 available binding sites, located in the twofold-symmetric interface connecting E2 core trimers. In *N. crassa*, the PX C-terminal domain oligomerizes into trimers. Oligomerization of PX limits binding stoichiometry to at most 24 PX monomers per E2 core, as a result of PDC core geometry (cf. Supplementary Fig. 1), and results in tetrahedral symmetry under favorable arrangement (Supplementary Fig. 1, config 8S). The size of the PX oligomers also limits the available configurations through volume occlusion, so that the *N. crassa* E2 core can accommodate at most 12 interior PX monomers. Saturation of such optimal arrangement of PX oligomers again result tetrahedral symmetry of the PDC core interior. Volume occlusion does however degenerate the optimal arrangement into two distinctly different, tetrahedral arrangements (Supplementary Fig. 1, config X4S and Y4S). We find ~60% of natively purified *N. crassa* PDC particles to be complete in this sense, but also observe directly that proportions of input data classified in conjunction with marginalizing reconstructions are not reliable indicators of the true distribution of states. Classes X and Y are however likely to dominate in the present sample, by evidence of the reemergence of sup-symmetric tetrahedral reconstructions under lower symmetries, and their correspondence to the geometrically optimal arrangements of interior density. Furthermore, threefold oligomers without occlusion should result in a tetrahedral arrangement that we see no evidence of. PX assembly and volume occlusion could also vary in related species, depending on the precise fold, oligomerization, or binding partners of interior PX components, such that other symmetries may be more prevalent in related fungal PDCs. We also note that under trimerization of PX each such trimer it is likely to coincide with the threefold symmetry of the E2 core assemble, which precludes it from being coined a symmetry mismatch. The global arrangement of multiple such trimers may however have arbitrarily low (a)symmetry.

N-terminal truncation of PX to include only the CBD has shown a binding of 30 PX per core in *S. cerevisiae*[23,27], consistent with monomeric PX binding. A reduction to 24 PX per core was observed if both the PSBD and CBD are left intact. Using our structural results, we can now attribute the latter to binding of a trimeric PX oligomer without volume occlusion as depicted in Supplementary Fig. 1g. Our observations of altered binding affinity or stoichiometry in PX301 and PX321 (Fig. 3c), as well as indications of a long-range beta-sheet interaction (based on bioinformatics analysis, Supplementary Fig. 7) corroborate that the full CBD is required for oligomerization, but not for binding. An earlier report showed that reconstituted PDC incorporating E1 and E3 bound only 12 PX per core[22], consistent with volume-occluding trimers as in *N. crassa*. Volume occlusion by the PX oligomer may thus be further reliant on E1 and/or E3 binding, at least in *S. cerevisiae*. This has been rationalized by way of E3 docking into the pentagonal faces of the E2 scaffold[23], a mechanism that seems unlikely given the PSBD of PX and the need for CoA to diffuse into the core interior. Moreover, we find no evidence of E3 docking directly to E2 in the native PDC from *N. crassa*. More likely, E1 and/or E3 might stabilize the PX trimer or contribute to its volume occlusion by binding to it. This might also explain why classification of recombinant E2–PX subcomplexes consistently fails to identify a tetrahedral core interior similar to the native *N. crassa* PDC in our hands, despite the obvious occupancy of PX in the core interior (Fig. 3a). While our reconstruction invites a direct interpretation of previous findings, further biochemical investigation of binding stoichiometry that considers it explicitly will be necessary to corroborate the structural regulation of PDC composition suggested. The discrepancy in sequence identity and conservation comparing the Pez and Sac-CBD (Supplementary Fig. 7) also merit future research to

completely understand previous findings. The presented mode of PX binding in *N.crassa* should greatly facilitate this.

If PX can bind in a monomeric form when oligomerization is disrupted[24], it remains unclear why oligomerization of the CBD with or without volume occlusion prevents binding of additional monomeric PX. We first consider that PX might be an obligate oligomer in its full-length form. This however seems unlikely, given the limited contacts made within the oligomer and the inability of PX oligomers to fit through the pentagonal openings of preformed core assemblies. It seems more likely that PX oligomerizes upon binding to the core. We also observe fast incorporation of PX30 into the PDC core during subcomplex reconstitution, further supporting a model in which PX oligomerization and binding are mutually enhancing. So what prevents monomeric PX from binding in excess of oligomerized PX as indicated by previous investigations? The identification of multiple similar conserved motifs in Pez consistent with the core binding pocket (Supplementary Fig. 7, motifs M1–3) could suggest that each PX monomer persistently occupies multiple binding sites; however, this does not explain previous observations in *S. cerevisiae*, since only the M3 motif (DJJDφLxG) is found in Sac. Instead, we suggest that oligomerization may be required for retained ability to bind when the dynamics of the extended N-terminal domain(s) are intact. This is appealing as a mechanism of PDC regulation through fold stability, and possible ligand and/or substrate interactions, and is consistent with all previous observations.

Finally, we note that ~40% of the endogenous PDC particles did not classify as containing either of the two tetrahedral arrangements of core-interior PX. This is most likely due to an unsaturated interior, but we cannot exclude the possibility of asymmetric interior due to suboptimal arrangements in these particles. Each PX monomer binds to an interface formed by multiple core trimers, so that PX can only bind partially or completely assembled cores. This agrees with the notion of monomeric PX that oligomerizes upon PX binding to access the core interior, and is supported by our observations of fast incorporation of PX into assembled E2 cores following separate expression. Interestingly, in a naive model of assembly in which trimeric PX oligomers attach sequentially to a preexisting core, only 15% would be expected to be in each tetrahedral arrangement—half of what is indicated by our classified proportions of cryo-EM particles. This discrepancy could possibly be attributed to uncertainty of classification due to possible data hiding. However, if the classification is accurate, it would suggest that PX co-assembles with the PDC core by binding to core fragments, which leads to an enrichment of optimal final arrangements and thus higher efficiency.

## Methods

**N. crassa cultivation and mitochondrial purification.** Frozen *N. crassa* (WT: STA74 / ATCC 14692, KO: FGSC 15821) mycelium was thawed and inoculated on 50 ml 2% agar minimal (Vogel's) medium supplemented with 1.5% sucrose and 1.5% Na-Acetate in 200 ml flasks. After incubation for 72 h at 30 °C in darkness followed by 72 h at room temperature in daylight, or after conidia appeared, conidia were harvested and filtered using sterile water. A total of 500 ml Vogel's medium supplemented as above were inoculated with conidial suspension, and grown in an flask (2 l) for 24 h at 25 °C under illumination and shaking at 140 r.p.m. A total of 25 ml of the resultant growth were then used to inoculate each of 20 E-flasks (2 l) containing 500 ml Vogel's medium, and then grown for 16 h at 25 °C under illumination and 140 r.p.m. shaking. Subsequent steps were conducted on ice, or at 4 °C. A totalm of 50 g of mycelium was collected by filtration through double-layer muslin, and ground using a chilled mortar, adding 100 g SiO₂ sand and 50 ml sucrose-EDTA-MOPS (SEM) buffer containing 0.5% phenylmethylsulfonyl fluoride. Sand was pelleted by centrifugation at 2000 r.c.f. for 10 min, collecting supernatant. The pelleted sand and cellular material mixture was ground twice more and pelleted, each time adding fresh buffer and pooling supernatant. Differential centrifugation then followed; 17,500 r.c.f. for 20 min, collecting pellet. The pellet was resuspended in 30 ml SEM and clarified at 2000 r.

c.f. for 10 min. Crude mitochondria were then pelleted at 17,500 r.c.f. for 20 min, and resuspended 10 ml SEM. The resuspended crude mitochondria were loaded on a 15–23–32–60% (4.5–4.5–12–4.5 ml) sucrose gradient and centrifuged in a Beckmann SW28 swing-out rotor at 100,000 r.c.f. for 1 h. The mitochondrial band was collected from the 32–60% interface and frozen at −80 °C.

**Endogenous PDC purification.** Mitochondria were thawed on ice and lysed using DDM to a final concentration of 2%, incubated under mild rocking for 15 min. Non-solubilized matter was pelleted and removed at 30,000 r.c.f. for 20 min. The clarified solution was loaded onto one or more 1 M sucrose cushions (7 ml) in Ti70 tubes, and the sample pelleted in a fixed-angle rotor using 230,000 r.c.f. for 4 h. Pelleted material was gently washed and resuspended in 0.4 ml of 50 mM HEPES-KOH (pH 7.5), 20 mM KCl, 1 mM DTT, then clarified at 15,000 r.c.f. for 10 min. The clarified solution was loaded onto a continuous 20–40% sucrose gradient in SW40 tubes and separated in a swing-out rotor at 70,000 r.c.f. over 16 h. PDC particles formed a visible band that started rather abruptly at 33% sucrose, extending with a smooth decrease toward higher sucrose concentrations. Fractions containing PDC were collected from the gradient using a long needle syringe. This produced pure and intact PDC particles within 20 h of mitochondrial lysis. Buffer exchange and purification was conducted in two steps. First, material was pelleted in TLA120.2 tubes at 100,000 r.c.f. for 2 h, resuspended in 0.25 ml fresh buffer, and twice clarified at 15,000 r.c.f. Finally, it was fractionated using size-exclusion chromatography (SEC; GE Superose 6-increase 3.2/100).

**Recombinant expression and purification.** The following genes were purchased from Life Technologies Europe for bacterial expression of *N. crassa* genes: E2 (29–458 uniprot:P20285), tE2 (225–458 uniprot:P20285), PX (30–426 uniprot: Q7RWS2). The pRSET vectors were used for single expression and provided a His6-tag that is cleavable using either thrombin (E2, tE2) or TEV (PX). petDuet dual-expression vectors were constructed using these as template, in all cases positioning the PX gene in the His-tagged MCS1. All vectors were amplified using *Escherichia coli* DH5-α, and all expression utilized Rosetta2 (DE3). For expression, cells were grown in terrific broth at 37 °C and 180 r.p.m. until OD reached ~0.5. Expression was induced by addition of IPTG to 1 mM and allowed to proceed for 3 h before harvest. Expression was consistently found to be worse if conducted for 16 h at 18 °C.

Cells were pelleted and resuspended in 50 mM imidazole buffer, then lysed either through sonication (small scale, e.g., truncation affinity assay) or high-pressure homogenization (large scale, cryo-EM preparations). Intact cells and debris were pelleted, and the supernatant collected. Ni-NTA agarose slurry was added and incubation under agitation proceeded for at least 30 min. Isolation and washing of Ni-NTA was performed either through repeated pelleting (small scale), or a gravity flow column. Depending on the downstream purpose, protein was either cleaved using TEV or eluted using imidazole. E2 was found to be well-expressed in all examined forms. Expression of PX was found to be much lower, but greatly improved in the presence of co-expressed E2 or tE2.

**Resolving the knockout homokaryon.** A NCU00050-knockout strain was purchased from the FGSC (15821), available as a heterokaryon with a wild-type helper genome to maintain viability on standard media. We resolved the KO homokaryon by generating monokaryotic spores on crossing media[35] with 2% sucrose and 1.5% agar. Microconidia were purified by filtering through double-layer 5V Whatman filters. The presence of microconidia and the absence of macroconidia was confirmed by microscopy. Spores were diluted appropriately and plated on fresh crossing media containing 40 mM acetate, 1% sorbose (to promote colonial growth), 1.5% agar, and 300 μM hygromycin (selecting for the knockout genome). The use of acetate in place of sucrose enabled a KO homokaryon to survive. Ten colonies were picked after a few days and replated on 5 ml fresh Vogel media plates, using 40 mM acetate, 1.5% agar, and 300 μM hygromycin, and grown until spores were produced. Spores were again purified and next subjected to a differential growth assay in Vogel's media, where a homokaryon knockout was expected to grow more favorably in 40 mM acetate compared to 40 mM sucrose. Cultures for two out of the ten colonies followed this expectation. The above procedure was repeated for one of these two cultures to ensure complete purity, supplementing the crossing media with 40 mM Na-acetate. The subsequent round of the differential growth assay showed ten out of ten colonies growing better on Na-acetate than sucrose. One of these colonies was used for subsequent cultivation and purification.

**Activity assay.** The activity of intact PDC assemblies was assayed by time-dependent spectroscopic monitoring of NADH production at 340 nm and room temperature. All samples for activity measurements were purified by SEC. The reaction was conducted in a UV-transparent (Sigma-Z628026) cuvette, with reactants as described in Supplementary Table 1.

All components apart from pyruvate and purified PDC were added and mixed, and a baseline was collected. Purified PDC was added at $t = −1$ min. At $t = 0$ min the reaction was initiated by addition of pyruvate. The activity decreased with time and was therefore resolved in time covering a 30 s sliding window, and linearly extrapolated to $t = 0$. At no stage of growth, purification or assay was phosphatase or dichloroacetate introduced to promote dephosphorylated E1.

**Mass spectrometry.** Peak fractions following SEC were immediately frozen and used for MS. Protein identification and quantification were carried out by the Proteomics Biomedicum core facility, Karolinska Institute. Analysis was performed using scaffold and scaffold PTM (posttranslational modification). PTM analysis was limited to known cofactors TPP and lipoamide, as well as phosphorylation.

**Cryo-EM grid preparation and data collection.** Grids for cryo-EM were prepared by glow discharge in a Pelco easiGlow. A total of 3 μl of sample was then applied to the grid and vitrified in a FEI Vitrobot mark IV, following 30 s wait, 2 s blot, and 2 s additional wait before plunging. A total of 100% humidity and 4 °C was maintained prior to plunging. Grid screening and optimization, as well as data collection was conducted at the Swedish National Cryo-EM Facility at SciLifeLab, Stockholm University and Umeå University. The parameters of grid preparation and data collection are summarized in Supplementary Table 2.

**Cryo-EM processing.** Preprocessing was conducted using motioncor2 (ref. [36]) and Gctf[37]. All reconstruction, alignment, classification and postprocessing was conducted using RELION[38]. The parameters of data processing and 3D reconstruction are summarized in Supplementary Table 3, and described in Supplementary Methods.

**Model building.** A molecular model of *N. crassa* E2 was built (PDB 6ZLO) into the map of the tE2 + PX30 reconstituted subcomplex (EMDB 11270), covering residues 227–458. This model was the basis for building a molecular model of the native PDC core, including the PX BH (PDB 6ZLM) into the reconstruction of endogenous PDC and interior in the S4Y configuration (EMDB 11268). Model validation metrics are shown in Supplementary Table 3. Molecular models were constructed using Coot[39], and refined using PHENIX[40]. Images were produced using UCSF Chimera[41] and ChimeraX[42].

**Bioinformatics.** Sequences annotated as E3BD containing were extracted from PFAM-32.0 (ref. [43]) and subjected to a preliminary multiple sequence alignment (MSA) in Jalview[44]. E2-like proteins were identified by presence of a conserved catalytic DHR-motif and/or preservation of secondary structure consistent with the known fold of E2, and omitted from further analysis. Alignment of this set of sequences formed the Ascomycota PX MSA. All retained sequences had been taxonomically annotated by PFAM as one of the five Ascomycota classes Sordariomycetes, Eurotiomycetes, Leotiomycetes, Dotiodeomycetes, or Saccharaomycetes. These are not true taxonomic classes, and are thus referred to as groups in the present work (Supplementary Fig. 7). The CBD was identified as everything following the flexible linker C-terminal to the E3BD-homologous domain, and isolated based on the PX MSA. Following a nearest-neighbor tree-clustering under BLOSUM62 substitution score, Pezisomycotina (Pez) and Saccharomycotina (Sac) were separated, and realigned individually, forming the Pez-CBD and Sac-CBD MSA, respectively (Supplementary Fig. 7c, d). Coevolutionary correlation was conducted through the raptorX[45] MSA-interface using the established CBD MSAs. We were unable to find any CBD-homologous sequences outside of Ascomycota.

## Data availability

All reconstructions presented have been deposited in the Electron Microscopy Data Bank, under EMDB accession codes 11266 (native icosahedral), 11267 (native tetrahedral config. X), 11268 (native tetrahedral config Y), 11269 (tE2 icosahedral), 11270 (tE2 + PX30 icosahedral), and 11271 (tE2/PX261 icosahedral). Atomic models have been deposited in the Protein Data Bank under PDB accession codes 6ZLM (E2 + PX-BH into EMDB 11268), and 6ZLO (E2 into EMDB 11270). The motion-corrected micrographs of the native preparation has been deposited in the EMPIAR database under accession code 10489, along with particle coordinates and angles determined for EMDB 11266, 11267, and 11268. Any additional data can be made available from the authors upon reasonable request.

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

## Acknowledgements

The presented research would not be possible without an initiative by Alexey Amunts, who we also thank for a lot of useful advice, suggestions, and encouragement throughout the project. The cryo-EM data were collected at the Swedish national cryo-EM facility, staffed by M. Carroni, J. M. de la Rosa Trevin, J. Conrad, and S. Fleischmann. We also thank the members of Amunts and Lindahl lab for active discussions throughout the project. This work was funded by the Swedish Foundation for Strategic Research (FFL15:0325), the Ragnar Söderberg Foundation (M44/16), the Swedish Research Council (2015-04107 and 2017-04641), Cancerfonden (2017/1041), EU grant ERC-2018-StG-805230 and BioExcel-823830, the Knut and Alice Wallenberg Foundation (2018.0080), and the Lennander Foundation. It was also supported by the Knut and Alice Wallenberg Foundation, Family Erling Persson Foundation, and Kempe Foundations through the Swedish National Cryo-EM Facility.

## Author contributions

B.O.F. designed experiments, collected and processed data, and wrote the article. S.A. designed experiments and collected data. R.J.H. designed experiments and wrote the article. N.M. contributed essential data and experimental procedures. E.L. supervised data processing and wrote the article.

## Funding

## Competing interests

The authors declare no competing interests.
