## [Peer Review File · Nature Communications]

Reviewer #1 (Remarks to the Author):

Review of "Assembly and symmetry of the fungal E3BP-containing core of the Pyruvate Dehydrogenase Complex"

This manuscript describes structural studies on a fungal version of the Pyruvate Dehydrogenase Multienzyme complex. Despite a number of studies on the complex and its many components, most from other species, its central role in metabolic processes, complexity, unusual architectures, and use of a target for numerous diseases makes it a prime candidate for additional studies as much remains unknown. The paper provides a lot of useful information on the fungal version, not as well studied as some of the other forms, where many questions also remain. I think a few issues need clarification to make for a stronger, and perhaps more generally appealing paper, as follows:

- 1) It would be useful to provide a sequence comparison between PX and E3BP, as it may give additional information to that in the Pez sequence included.
- 2) pg 2, 2nd par. Tethering (PSBD) linkages may not be badly disordered. It's more likely the LD segments that MUST be disordered, as their function actually requires large motions, and there is nothing present to keep them anchored at any of the active sites. It may just be faulty class assignments prior to averaging that yield apparent disorder in the PSBD linkages, since they remain static throughout and also span much shorter distances..
- 3) The statement "Finer control over metabolic flux is also afforded by adjusting the relative proportions of the PDC components, regulating its overall activity" is ambiguous. Is this suggesting a DYNAMIC change in composition during the proteins functioning, or a static change that differs in species (or tissue) and is maintained throughout?
- 4) Regarding statements about E3BP in some species and PX, I also note that CORE component E3 binding proteins rather than E2 still have viable lipoyl domains, and could still participate in ALL acetyl group transfer related activities, i.e. from pyruvate to lipoamide in E1, acetyl transfer to CoA in E2, and dithiolane ring closure in E3. They would just have to deliver the acetyl group to a NEIGHBORING (likely adjacent) E2 active site for transfer to CoA rather than to its own, dead, E3BP catalytic site. As long as the viable E2 active site is readable, distance wise, this should still be possible. Whether this has any bearing on PX though, is unknown.
- 5) Note that component distributions in PDC's having symmetries inconsistent with that of the apparent, overall complex is not unusual. As examples, the stoichiometry in mammalian versions violate proper icosahedral symmetry when the regulatory components (and even E3BP) are included, and in the bacterial PDC's having apparent cubic symmetry it's actually octahedral as the E3 dimeric components can NOT have 4-fold symmetry on each face (there aren't enough of them). Perhaps it would be useful to point this out.

Reviewer #2 (Remarks to the Author):

The manuscript by Forsberg et al. characterises the PX component of the fungal PDC. To achieve this, the authors study three sample types: fully assembled native PDC (E1-E2-PX-E3), recombinant PDC subcomplex (E2-PX) and recombinant E2. The work focuses on determining cryo-EM reconstructions for these samples. 60 copies of E2 form the icosahedral core of the PDC while the other components deviate from icosahedral symmetry. As the cryo-EM reconstructions of E1-E2-PX-E3 and E2-PX present similar density on the interior of the E2 core which is absent from E2 alone, this interior density is assigned to PX. Further cryo-EM image processing discerns two configurations of PX, both displaying tetrahedral symmetry which suggests the presence of 12 PX monomers. Two neighbouring

regions of PX density are identified in the proximity of E2 and denoted as the (E2-)binding helix (BH) and the oligomer domain (OD), respectively. The resolution of BH and OD is not sufficient to trace the protein sequence. In order to identify the PX sequence motifs corresponding to BH and OD, the authors perform multiple sequence alignments of PX as well as E2-PX binding studies. The findings are presented in the context of previously suggested E2:PX stoichiometries in fungi and the fungal PDC is contrasted to the mammalian PDC. Ultimately, the performance and limitations of the employed cryo-EM image processing methodology are discussed from a technical point of view.

In summary, the manuscript describes a novel tetrahedrally-symmetric structure of PX which confers a unique organisation to the fungal PDC compared to other known PDC structures. The choice of methodologies is sound and the results are largely well-presented. However, I have a number of remarks in regards to the cryo-EM image processing, interpretation of results and use of language throughout the text.

Major comments:

- The current title "Assembly and symmetry of the fungal E3BP-containing core of the Pyruvate Dehydrogenase Complex" appears to refer to the mechanism of assembly of the PDC, which is not discussed thoroughly enough to warrant its incorporation into the title.
- The PDC structures determined here provide valuable insights into the evolution of this fundamental molecular machine. However, there are a number of aspects related to cryo-EM that may cast doubt on the validity of the structures if left unaddressed.
- The authors state that the tE2+PX30 dataset gave rise to the same two tetrahedral PX configurations (X and Y) as the native dataset did but to higher resolutions (3.2 Å and 3.6 Å compared to 4.2 Å and 4.3 Å, respectively, as described in Methods) and that "the improvement is localized to the E2 region". Later, the authors note at the end of paragraph 1 on page 10: "This might also explain why classification of recombinant E2-PX subcomplexes consistently fails to identify a tetrahedral core interior similar to the native *N. crassa* PDC in our hands, despite the obvious occupancy of PX in the core interior." Isn't this "failed" classification described both in Methods 3.8.2 as well as in Figure S6? If there are indeed tetrahedral reconstructions at 3.2 Å and 3.6 Å of E2-PX, one would expect that the resolution of the binding helix should be very similar to these. Isn't this the case?
- In order to improve the orientational search for (and hence the resolution of) the PX component during refinement, one could remove the contribution of E2 from the original cryo-EM images by partial signal subtraction in RELION. This has been shown to provide significant improvements for a number of lower symmetry specimens attached to highly symmetrical frameworks and I would expect that this would be the case here as well, at least for the E2+PX samples.
- Was the tE2/PX261 dataset subjected to the same image processing procedures as native and tE2+PX30? There is no mention of any processing for tE2/PX261 in Methods. I would expect tE2/PX261 to provide the structures with the highest overall resolution for PX as PX261 has a much higher proportion of its structure bound to E2 compared to PX30.
- How did the authors make the decision to apply tetrahedral symmetry during image processing? Indeed the authors state on multiple occasions that the symmetry of the complex is "pseudo-tetrahedral" despite applying tetrahedral symmetry in their reconstructions. Applying a wrong symmetry in cryo-EM processing leads to incorrect averaging and hence limits the resolution of the reconstruction, as the authors point out in the case of applying icosahedral symmetry to PX. Have the authors attempted to obtain an asymmetric (i.e. C1) reconstruction of PX? If PX is indeed tetrahedral, the reconstruction obtained by classifying/refining the structure without any symmetry imposed should also appear tetrahedral. Another alternative would be to perform localized reconstruction and classification of the 20 individual E2 vertices in order to discern the ones bound to PX from the ones free of PX. One could then trace these individual E2 vertices back onto their original assemblies in order to determine the overall organisation of PX.
- The representation and interpretation of Figure S9 are ambiguous. As the authors only show slices through the 3D reconstructions as well as "a region of interest with increased contrast", it remains unclear how much information can be gained from their analysis. First, class X in c) is very similar to class X in d) (though the inset appears to be shifted between the two), whereas the inset of class Y in c) is different from the inset of class Y in d) which appears to begin resembling a) and b). Second, the authors state that the left panel in f) resembles class Y in d) though the inset appears to also start

resembling a) and b) as above. Third, the authors state that "the superimposed class X and Y is then also recovered in line with expectation"; however, the inset of the left panel in f) appears most similar to the inset of class X in d).

- At times, I find the language of the paper to be too technical for the general readership of Nature Communications. The authors use jargon related to cryo-EM image processing without the elaboration required for understanding by a reader without expertise in the field. Some examples:

- In paragraph 2 of Introduction, the authors dive abruptly from describing the components of the PDC into image reconstruction theory that is not easily understandable even for a prototypical cryo-EM user.

- In paragraph 3 of Introduction, the authors introduce the concept of "implicit symmetrisation." While this does not necessarily represent a problem, I believe the authors should discuss the issue of samples containing a mixture of symmetries in light of "symmetry mismatches", a much more commonly used term in the field (introduced by Hendrix 1978, reviewed recently by Huiskonen 2018 and Goetschius et al. 2019). I strongly recommend that the authors write a separate paragraph in Introduction briefly describing symmetry mismatches and the issues they pose in determining the cryo-EM structure of the fungal PDC. In fact, the authors use the term "cryo-EM" for the first time only in the very last line of Introduction, which further warrants the need for the additional paragraph.

- The authors should explain the mathematical term "marginalize" which they use on a number of occasions i.e. "marginalized reconstruction", "marginalized classification", "ability to marginalize", "extent of marginalization" etc. This being said, most of section 1.5 is nonetheless rather hard to digest.

Minor comments:

- Have the authors checked whether any of the PX constructs can oligomerise in the absence of E2? This observation could help clarify their discussion of the PDC assembly.

- The abstract is too vague. The authors mention that the fungal PDC is structurally distinct from the mammalian one and that their results explain previous observations in fungi. Yet, they do not comment on the structure of the mammalian PDC or the previous observations in fungi.

- Most of paragraph 2 of Introduction is missing references.

- In Figure 1a) the authors refer to the image as "Reconstruction of the intact native PDC icosahedral Asymmetric unit (I-ASU)". What they depict however is the entire icosahedral assembly (i.e. composed of 60 asymmetric units). This should be changed throughout the manuscript when referring to I-ASU and T-ASU as they discuss the whole assembly rather than just an asymmetric unit of the assembly.

- Explanations for "positive direction" and "negative direction" of a symmetry axis, as well as for "affine transformation" are needed.

- The authors mention "weighted backprojection" in their discussion of image processing. Does RELION actually employ this procedure?

- Typos: "explaining previous observation", "aerobic glycolysis", "phosphorylation of E1 by kinases and phosphatases", "withinth"

Reviewed by Serban Ilca

Reviewer #3 (Remarks to the Author):

The authors have made an extensive effort to characterize the inner core structure of *N. crassa* pyruvate dehydrogenase complex (PDC) that is formed by the C-terminal oligomer-forming, domains of the E2 and protein X (PX) components (here E2i and PXi). E2i forms a typical 60 domain icosahedral dodecahedron structure with 20 trimers at the corners and 12 pentagonal faces via 2-fold interactions among trimers. Like mammalian PDC, *N. crassa* PX is shown to bind E3 and this retention of E3 is required for PDC function. Similar to *S. cerevisiae* PDC inner core, the PXi domain is shown to be held inside the E2i dodecahedron; this is in marked contrast to mammalian (human best characterized) PDC inner core in which E3BP inner domains substitute for the related E2i domains within the dodecahedron. The authors effectively show that PXi form trimers and binding of 4 PXi

trimers within the dodecahedron employs a different symmetry (tetrahedral rather than icosahedral) and that this remains the case when linker region held outer domains (lipoyl. and subunit binding) are removed by truncation. Likely requirements for gaining and retaining PXi domains bound to E2i oligomer are discussed (more below). The studies are thorough, well performed and a substantial advance.

Most of the following review analysis involves requesting clarifications and deserve responses.

While the essential role of PX for retention of E3 with the complex is established, it would seem that a requirement of PX for some PDC function is not since studies were not conducted with excess E3 added to complexes lacking PX (PX knockout preparations).

A tetrahedral arrangement involving 4 trimers of PXi with equivalent physical positioning is indicated. This is implicitly indicated but never explicitly stated which would aid a nonexpert readers. Besides physical constraints on space and the nature of interactions of PXi trimer subunits with E2i at 2-fold axis within its icosahedral structure, the presence, absence, or unknown status of any interaction between trimers needs to be explicitly addressed. The likelihood for such interactions could be exhibited more clearly with removal of E2i dodecahedron (leaving 4 trimers) along with insertion of a tetrahedral diagram (assume corners of which would be at central holes of trimers). Only part of the PXi domain structure was resolved; the possibility that nonresolved parts of the domain trimers could participate in inter trimer interactions that help maintain the tetramer arrangement should be part of this evaluation.

Indications of motion of bound trimers are not indicated even though the E2i-binding of BH segment of Pxi at 2-fold axes appear mobile as considered in the Discussion. The degree to which the involvement of 3BH binding per trimer could limit this motion should be considered and, if likely, role inter-trimer interactions. Breathing (expansion/contraction) was found in studies of other inner core E2 oligomers with the primary change along 2-fold axes. For comparison, a critical experimental step is not described in this work. What temperature was the grid and E2 or E2-PX complexes at prior to being rapidly frozen? Expansion/contraction of inner cores is only detected with the T elevated (e.g. RT) and not with low temperature solutions of components (e.g. ice-maintained temperatures).

Similarly, the temperature at which PDC activity assays are conducted is not described. Is it possible that the low activity of complexes is due to exposure in cells/mitochondria to pyruvate (derived from sucrose) along with TPP but in the absence of non-acetylated CoA which leads to inactivation of E1 with most sources of complex. This is why adding enzyme or pyruvate last is important in performing activity assays.

Minor:

In Figure 3b, the binding domain for E2 should be labeled E1BD not E3BD.

The acidic/hydrophobic M3 is flanked by classic linker region sequences but closest in by runs of lysine residues. Assuming this segment does play the BH role as suggested, is any role proposed for these residues and what would prevent these basic residues from disrupting interactions of M3 with basic residues in 2-fold E2i domain bridges?

Indicate reference (pdb source) for Fig. 2d. Reference 18 does indicate a distinct protein (called protein X), but this source provides no evidence for E3 binding role. A reference to this discovery is needed.

Despite the mammoth amount of sequence data conveyed in Figure S7, it is not clear to this nonexpert, the degree to which the findings extend to other fungi. There appear to be 13 classes of Pezizomycotina (wiki). Are the three (besides Sordariomycetes which includes Neurospora) broadly representative of classes or particularly close to Sordariomycetes? Clearly Saccharomycetes are

substantially different as are the undefined "Various" fungi (include Taphrinomycotina?).

Comments:

This reviewer agrees with the Discussion comments indicating that assembly of oligomers of PXi domains occurs within the dodecahedron since trimers could not fit through the five-fold faces and that assembly is probably required for retaining PXi domains in this space

While agreeing that sequences of *S. cerevisiae* and *N. crassa* PXi are at best distantly related (Fig. 7S), the authors are generous regarding the putative findings of prior studies with *S. cerevisiae* PXi binding in larger amounts to E2i oligomer. Cryo-EM results were only analyzed using icosahedral symmetry (possibly giving misleading results like Fig. 1a) and the 1996 paper (ref. 24) suggesting higher stoichiometric binding of truncated PXi does not provide extant supporting data with the many needed controls, but it only supplies a minimal Table that must be accepted based on faith. These limitations could be indicated.

Erik Lindahl
Professor of Biophysics

Response to reviewer comments - NCOMMS-20-14293-T

Dear Dr. Marcinkiewicz,

Thank you very much for the positive and highly constructive comments from all three reviewers. We believe we have been able to address all of them and have updated the MS accordingly. Our responses to the comments below is in blue italics.

Reviewer #1:

This manuscript describes structural studies on a fungal version of the Pyruvate Dehydrogenase Multienzyme complex. Despite a number of studies on the complex and its many components, most from other species, its central role in metabolic processes, complexity, unusual architectures, and use of a target for numerous diseases makes it a prime candidate for additional studies as much remains unknown. The paper provides a lot of useful information on the fungal version, not as well studied as some of the other forms, where many questions also remain. I think a few issues need clarification to make for a stronger, and perhaps more generally appealing paper, as follows:

1.1 It would be useful to provide a sequence comparison between PX and E3BP, as it may give additional information to that in the Pez sequence included.

Good idea. We have included a sequence comparison of human and fungal PX as supplementary figure S8, which is referred to in the text. Moreover, the limited sequence similarity of fungal and mammalian PX/E3BP (Fig. S8) led us to analyse the fungal CBD without consideration for mammalian E3BP sequence data.

1.2 pg 2, 2nd par. Tethering (PSBD) linkages may not be badly disordered. It's more likely the LD segments that MUST be disordered, as their function actually requires large motions, and there is nothing present to keep them anchored at any of the active sites. It may just be faulty class assignments prior to averaging that yield apparent disorder in the PSBD linkages, since they remain static throughout and also span much shorter distances.

Agreed. It is quite correct that PSBD linkages are not necessarily disordered, but this was not clearly indicated in the MS - we have clarified this in the text (see below). Additionally, our interpretation of the data assumes that ordered PSBD-linkers would result in E3 being ordered in N. crassa by consequence of the ordered PX, which we observe no evidence of. This indicates disorder or flexibility in the OD-PSBD linking region, which is also evidenced by low sequence conservation (Fig S7). The reviewer is also right this linking region is fairly short in the MSA (Fig S7) compared to the ~60AA of unannotated linker region in fungal PX. The annotation of the domain structure inferred in Fig S7A has been amended to better indicate that part of the conserved sequence is outside the consensus PSBD-motif, and that the flexibility expected in the linking region of low sequence conservation is not assured. This has also been reflected in the text, stating in the introduction that "The tethering of E1 and E3 to the E2 core by disordered linking regions that are likely flexible and arranged less symmetrically than the core makes it challenging to reconstruct faithfully." We also remark in the results that "Based on sequence conservation, the PSBD seems to extend C-terminally compared to e.g. human E3BD, leaving only a short linker evidenced by low sequence conservation (Fig. S7). This might suggest increased order of E3 in fungi, but we observe no quantifiable order of the PDC periphery in our reconstructions to corroborate this".

1.3 The statement "Finer control over metabolic flux is also afforded by adjusting the relative proportions of the PDC components, regulating its overall activity" is ambiguous. Is this suggesting a DYNAMIC

Department of Biochemistry and Biophysics

Stockholm University
Science for Life Laboratory
Box 1031
SE-171 21 Solna, Sweden

Visiting address:
Science for Life Laboratory
Tomtebodavägen 23A, Solna

Phone: +46-734618050
Cell: +46-734618050
Mail: erik.lindahl@dbb.su.se

Erik Lindahl
Professor of Biophysics

change in composition during the proteins functioning, or a static change that differs in species (or tissue) and is maintained throughout?

*That formulation was indeed not very clear. It has been changed to “**The saturation and relative proportions of tethered E1 and E3 may also be subject to change in response to external cues and alteration in metabolic requirements.**”*

1.4 Regarding statements about E3BP in some species and PX, I also note that CORE component E3 binding proteins rather than E2 still have viable lipoyl domains, and could still participate in ALL acetyl group transfer related activities, i.e. from pyruvate to lipoamide in E1, acetyl transfer to CoA in E2, and dithiolane ring closure in E3. They would just have to deliver the acetyl group to a NEIGHBORING (likely adjacent) E2 active site for transfer to CoA rather than to it's own, dead, E3BP catalytic site. As long as the viable E2 active site is readable, distance wise, this should still be possible. Whether this has any bearing on PX though, is unknown.

*Agreed. The catalytic inactivity of E3BP in mammals does not prevent PDC function, and its carrier domains may still be functional and utilise adjacent E2 catalytic sites. We have clarified this by modifying the statement to “**In mammals, this is further exploited through the expression of a catalytically inactive E2 variant with viable substrate-carrier domains [Demarcucci1985]...**”*

1.5 Note that component distributions in PDCs having symmetries inconsistent with that of the apparent, overall complex is not unusual. As examples, the stoichiometry in mammalian versions violate proper icosahedral symmetry when the regulatory components (and even E3BP) are included, and in the bacterial PDCs having apparent cubic symmetry it's actually octahedral as the E3 dimeric components can NOT have 4-fold symmetry on each face (there aren't enough of them). Perhaps it would be useful to point this out.

*Good point. We have clarified that symmetry mismatches are not uncommon in PDC from various taxa, and that the fungal PDC is not unique in this respect: “**The challenge to determine how this regulation is dictated by structure lies in the suppression of relatively small asymmetric features by a largely symmetric complex, here termed implicit symmetrization. This is a topic of ongoing research, largely focused on coinciding but incompatible symmetries [Huiskonen2018], termed symmetry-mismatches**”.*

Reviewer #2:

The manuscript by Forsberg et al. characterises the PX component of the fungal PDC. To achieve this, the authors study three sample types: fully assembled native PDC (E1-E2-PX-E3), recombinant PDC subcomplex (E2-PX) and recombinant E2. The work focuses on determining cryo-EM reconstructions for these samples. 60 copies of E2 form the icosahedral core of the PDC while the other components deviate from icosahedral symmetry. As the cryo-EM reconstructions of E1-E2-PX-E3 and E2-PX present similar density on the interior of the E2 core which is absent from E2 alone, this interior density is assigned to PX. Further cryo-EM image processing discerns two configurations of PX, both displaying tetrahedral symmetry which suggests the presence of 12 PX monomers. Two neighbouring regions of PX density are identified in the proximity of E2 and denoted as the (E2-)binding helix (BH) and the oligomer domain (OD), respectively. The resolution of BH and OD is not sufficient to trace the protein sequence. In order to identify the PX sequence motifs corresponding to BH and OD, the authors perform multiple sequence alignments of PX as well as E2-PX binding studies. The findings are presented in the context of previously suggested E2:PX stoichiometries in fungi and the fungal PDC is contrasted to the mammalian PDC. Ultimately, the performance and limitations of the employed cryo-EM image processing methodology are discussed from a technical point of view.

In summary, the manuscript describes a novel tetrahedrally-symmetric structure of PX which confers a unique organisation to the fungal PDC compared to other known PDC structures. The choice of

Erik Lindahl
Professor of Biophysics

methodologies is sound and the results are largely well-presented. However, I have a number of remarks in regards to the cryo-EM image processing, interpretation of results and use of language throughout the text.

2.1 The current title "Assembly and symmetry of the fungal E3BP-containing core of the Pyruvate Dehydrogenase Complex" appears to refer to the mechanism of assembly of the PDC, which is not discussed thoroughly enough to warrant its incorporation into the title.

This is a very good point. Assembly does imply temporal dependence, whereas we intended the architecture of the complex to be highlighted. We have changed the title from "Assembly" to "Arrangement", which is also used throughout the text now.

2.2 The PDC structures determined here provide valuable insights into the evolution of this fundamental molecular machine. However, there are a number of aspects related to cryo-EM that may cast doubt on the validity of the structures if left unaddressed.

These were indeed good ideas, and we have updated the MS accordingly.

2.2.1 The authors state that the tE2+PX30 dataset gave rise to the same two tetrahedral PX configurations (X and Y) as the native dataset did but to higher resolutions (3.2 Å and 3.6 Å compared to 4.2 Å and 4.3 Å, respectively, as described in Methods) and that "the improvement is localized to the E2 region". Later, the authors note at the end of paragraph 1 on page 10: "This might also explain why classification of recombinant E2-PX subcomplexes consistently fails to identify a tetrahedral core interior similar to the native *N. crassa* PDC in our hands, despite the obvious occupancy of PX in the core interior." Isn't this "failed" classification described both in Methods 3.8.2 as well as in Figure S6? If there are indeed tetrahedral reconstructions at 3.2 Å and 3.6 Å of E2-PX, one would expect that the resolution of the binding helix should be very similar to these. Isn't this the case?

It is not our intention to claim that the recombinant constructs achieve the same tetrahedral packing as observed in the native preparation, nor did we observe this, as mentioned in the second paragraph of the discussion. This could be due to any number of factors regulating how PX incorporates into the core endogenously, which we did not investigate in the present work. Rather, we note that the PX-component will always be over-symmetrized under icosahedral symmetry, and since the native and recombinant preparations are highly similar under icosahedral symmetry, we argue this supports the presence of PX bound interior to the PDC core in the recombinant preparations too. The statement regarding improved resolution in the recombinant preparations has been amended to clarify these aspects and clarify that tetrahedral arrangement of PX is only observed in the native preparation: "The overall resolution is better, likely due to the absence of the PDC exterior. However, we are not able to find a tetrahedral class of particles in this or any recombinant preparation, which we attribute to a decreased occupancy and order of the interior assembly compared to the endogenous preparation."

2.2.2 In order to improve the orientational search for (and hence the resolution of) the PX component during refinement, one could remove the contribution of E2 from the original cryo-EM images by partial signal subtraction in RELION. This has been shown to provide significant improvements for a number of lower symmetry specimens attached to highly symmetrical frameworks and I would expect that this would be the case here as well, at least for the E2+PX samples.

Very good point. In fact, we actually did attempt such an analysis, but unfortunately without observing any benefit - we now make sure this is mentioned in the text. Since the resolution of the E2 component of the core is much higher than that of the interior PX component, we take this to indicate that flexibility of PX rather than angular assignment accuracy is limiting resolution. Subtraction of the core leaves a small-mass PX that is only tetrahedrally ordered in the native preparation, where the periphery remains and supplies a high degree of noise. Individual alignment of PX-trimers to account

Erik Lindahl
Professor of Biophysics

for flexibility is thus difficult, which is how we rationalize the inability of core-subtraction to improve PX-density. We agree that this reasoning adds value to the analysis, and it has been added to section 1.2: “We also attempted to subtract the core and classify only the interior PX density. However, the periphery and low order of the endogenous and recombinant preparations, respectively, prevented this approach from supplying any additional information regarding the fungal PDC interior.”

2.2.3 Was the tE2/PX261 dataset subjected to the same image processing procedures as native and tE2+PX30? There is no mention of any processing for tE2/PX261 in Methods. I would expect tE2/PX261 to provide the structures with the highest overall resolution for PX as PX261 has a much higher proportion of its structure bound to E2 compared to PX30.

As tE2+PX30, the tE2/PX261 preparation failed to identify a homogenous class of tetrahedrally arranged core-interior PX. This means that while circumstances are otherwise optimal, PX is not better resolved. The similarity of tE2+PX30 and tE2/PX261 under icosahedral (over-)symmetry can also be seen in Fig. 3A. The methods section has been updated to reflect this, and we now also describe the tE2/PX261 processing there.

2.2.4 How did the authors make the decision to apply tetrahedral symmetry during image processing? Indeed the authors state on multiple occasions that the symmetry of the complex is "pseudo-tetrahedral" despite applying tetrahedral symmetry in their reconstructions. Applying a wrong symmetry in cryo-EM processing leads to incorrect averaging and hence limits the resolution of the reconstruction, as the authors point out in the case of applying icosahedral symmetry to PX. Have the authors attempted to obtain an asymmetric (i.e. C1) reconstruction of PX? If PX is indeed tetrahedral, the reconstruction obtained by classifying/refining the structure without any symmetry imposed should also appear tetrahedral. Another alternative would be to perform localized reconstruction and classification of the 20 individual E2 vertices in order to discern the ones bound to PX from the ones free of PX. One could then trace these individual E2 vertices back onto their original assemblies in order to determine the overall organisation of PX.

As mentioned in the first paragraph of the results section, all icosahedral sub-symmetries were attempted, but we have now clarified this further: “All icosahedral symmetries were attempted. While this alone did not aid the interpretation of the interior density for any such sub-symmetry, 3D classification coupled with the application of tetrahedral symmetry yielded a correctly averaged reconstruction symmetry, ...”. The “pseudo-tetrahedral” constructions were observed when even lower symmetries (tetrahedral sub-symmetries) were applied, e.g. D2 or C3. In these cases, classification identified reconstructions which depict a tetrahedral arrangement of PX, but without strict symmetry since only a sub-symmetry was enforced. This is what we term pseudo-tetrahedral. This use of terminology is perhaps inappropriate for its purpose, so we have changed the MS to rather use the terms sub-symmetry and sup-symmetry (see also response to reviewer comment 3.2.3 below), to be consistent with the nomenclature instated by Huiskonen2018.

2.2.5 The representation and interpretation of Figure S9 are ambiguous. As the authors only show slices through the 3D reconstructions as well as "a region of interest with increased contrast", it remains unclear how much information can be gained from their analysis.

Fair point. We do not think there is any obvious perfect way of conveying all the information, and volume slices offer the, in our opinion, most distinguishing view of class X and Y, as 3D volume views would obscure the interior and depend on the chosen threshold. Still, we acknowledge the concern and have amended the figure in response to the reviewers specific concerns below.

Stockholm University

Erik Lindahl
Professor of Biophysics

2.2.5a First, class X in c) is very similar to class X in d) (though the inset appears to be shifted between the two), whereas the inset of class Y in c) is different from the inset of class Y in d) which appears to begin resembling a) and b).

The figure [now S10] was included to convey that X and Y are distinctly different from each other and the remaining (possibly over-symmetrized) class of particles, and that a subsequent refinement of each class separately preserves these differences. As particle weights distribute across classes this is not given, and this also explains the minor differences noted by the reviewer. The reviewer is correct in pointing out that the insets represent an arbitrary point of detail that does not aid in making this general point. For this reason, the insets have been removed to avoid confusing the reader.

2.2.5b Second, the authors state that the left panel in f) resembles class Y in d) though the inset appears to also start resembling a) and b) as above.

We agree the inset alone could invite this interpretation, but maintain that the overall volume slice shows a clear similarity to d). As stated above, the insets have been removed to not cause a misleading emphasis on details, but rather show the overall (dis)similarity of the reconstructions.

2.2.5c Third, the authors state that "the superimposed class X and Y is then also recovered in line with expectation"; however, the inset of the left panel in f) appears most similar to the inset of class X in d).

The inset is perhaps not indicative in this case. As stated above, the insets have been removed for clarity. We have also added remarks to the effect that the right of panel f) cannot be expected to resemble any other reconstruction, as it is a mixture of distinct arrangements utilizing alignments determined for either arrangement separately, and hence is not a mode of over-symmetrization that would be expected from "conventional" processing. This is fact part of the point being made - that the reconstruction procedure does not represent mixtures in the way one might expect.

2.3 At times, I find the language of the paper to be too technical for the general readership of Nature Communications. The authors use jargon related to cryo-EM image processing without the elaboration required for understanding by a reader without expertise in the field. Some examples:

2.3.1 In paragraph 2 of Introduction, the authors dive abruptly from describing the components of the PDC into image reconstruction theory that is not easily understandable even for a prototypical cryo-EM user.

In hindsight, we agree with the reviewer; we have now adapted the description throughout the paper to suit a more general audience.

2.3.2 In paragraph 3 of Introduction, the authors introduce the concept of "implicit symmetrisation." While this does not necessarily represent a problem, I believe the authors should discuss the issue of samples containing a mixture of symmetries in light of "symmetry mismatches", a much more commonly used term in the field (introduced by Hendrix 1978, reviewed recently by Huiskonen 2018 and Goetschius et al. 2019). I strongly recommend that the authors write a separate paragraph in Introduction briefly describing symmetry mismatches and the issues they pose in determining the cryo-EM structure of the fungal PDC. In fact, the authors use the term "cryo-EM" for the first time only in the very last line of Introduction, which further warrants the need for the additional paragraph.

The reviewer raises an important consideration of nomenclature, also addressed in response to reviewer concern 2.2.4 and 1.5. The references indicated by the reviewer establish a "symmetry-mismatch" in the context of multiple (incompatible) symmetries specified around the same axis. Our reconstruction shows no indication of this, apart from C1-asymmetry (of the PDC periphery, which is not the topic of focus). Coining the sub-symmetric PX-arrangement within the core interior a

Erik Lindahl
Professor of Biophysics

*“symmetry-mismatch” does not exactly reflect our understanding of the data, and discussing it as such is possibly confusing. Rather, the core and interior together form a strictly tetrahedral reconstruction. It would e.g. be misleading to put the fungal PDC tetrahedral arrangement in the nomenclature of Huiskonen2018, since this does not make it evident that the PX-arrangement is itself tetrahedral. It is nevertheless useful to make the relation to symmetry-mismatches explicit, as suggested by the reviewer. To this end, we have amended the introduction, including the statement **“The challenge to determine how this regulation is dictated by structure lies in the suppression of relatively small asymmetric features by a largely symmetric complex, here termed implicit symmetrization. This is a topic of ongoing research, largely focused on coinciding but incompatible symmetries [Huiskonen2018], termed symmetry-mismatches.”** Moreover, the discussion elaborates on the role of symmetry-mismatches in the present context: **“We also note that under trimerization of PX each such trimer is likely to coincide with the three-fold symmetry of the E2 core assemble, which precludes it from being coined a symmetry-mismatch. The global arrangement of multiple such trimers may however have arbitrarily low (a)symmetry”.***

2.3.3 The authors should explain the mathematical term "marginalize" which they use on a number of occasions i.e. "marginalized reconstruction", "marginalized classification", "ability to marginalize", "extent of marginalization" etc. This being said, most of section 1.5 is nonetheless rather hard to digest.

*We agree that the terminology deserves to be better introduced, and now introduce marginalization by stating **“Strictly enforcing a symmetry makes any sub-symmetry over-symmetrized. Sup(er)-symmetric reconstructions can occur, wherein a higher symmetry than specified is reconstructed. This is most obviously the case when a partial sup-symmetry is present, e.g. the icosahedral core around the tetrahedral PX interior. This effect is enhanced when multiple orientations are utilized in a weighted manner for the backprojection of each particle, known as marginalization of the alignment. This leads to implicit symmetrization, which may obscure lower symmetries in favour of sup-symmetries. For this reason, specifying icosahedral sub-symmetries is not a rigorous method to classify PDC particles according to interior arrangement, despite its success in identifying two tetrahedral arrangements. In fact, no good method exists to classify particles according to symmetry.”** We have also tried to make section 1.5 more accessible to a general audience with interest in solving similar issues, i.e. interpreting and reconciling a sub-symmetric or asymmetric portion of cryo-EM data by marginalizing tools for classification.*

Minor comments:

2.4 Have the authors checked whether any of the PX constructs can oligomerise in the absence of E2? This observation could help clarify their discussion of the PDC assembly.

*PX does **not** appear to form a stable oligomer in solution. This statement has been added to section 1.2. The ability of PX to bind pre-assembled tE2 also indicates that oligomerization occurs only upon binding to core E2, since a PX trimer could not pass into the core interior through the pentagonal face opening. While we do not observe PX oligomers directly in these recombinant preparations, the over-symmetrized reconstructions shown in figure 3 clearly show that PX is ordered (but not symmetrically arranged) within the core interior, indicating oligomerization.*

2.5 The abstract is too vague. The authors mention that the fungal PDC is structurally distinct from the mammalian one and that their results explain previous observations in fungi. Yet, they do not comment on the structure of the mammalian PDC or the previous observations in fungi.

The abstract has been changed accordingly:

The pyruvate dehydrogenase complex (PDC) is a multi-enzyme complex central to aerobic respiration, connecting glycolysis to mitochondrial oxidation of pyruvate. Similar to the E3-binding protein (E3BP) of mammalian PDC, PX selectively recruits E3 to the fungal PDC, but its

Erik Lindahl
Professor of Biophysics

divergent sequence suggests a distinct structural mechanism. Here, we report reconstructions of PDC from the filamentous fungus Neurospora crassa by cryo-electron microscopy, where we found protein X (PX) interior to the PDC core as opposed to substituting E2 core subunits as in mammals. Steric occlusion limits PX binding, resulting in predominantly tetrahedral symmetry, explaining previous observations in Saccharomyces cerevisiae. The PX-binding site is conserved in (and specific to) fungi, and complements possible C-terminal binding motifs in PX that are absent in mammalian E3BP. Consideration of multiple symmetries thus reveals a differential structural basis for E3BP-like function in fungal PDC.

2.6 Most of paragraph 2 of Introduction is missing references.

Suitable references have been added, and the introduction was restructured in response to other reviewer concerns.

2.7 In Figure 1a) the authors refer to the image as "Reconstruction of the intact native PDC icosahedral Asymmetric unit (I-ASU)". What they depict however is the entire icosahedral assembly (i.e. composed of 60 asymmetric units). This should be changed throughout the manuscript when referring to I-ASU and T-ASU as they discuss the whole assembly rather than just an asymmetric unit of the assembly.

We have duly changed relevant passages, removing the use of I/T-ASU.

2.8 Explanations for "positive direction" and "negative direction" of a symmetry axis, as well as for "affine transformation" are needed.

We have clarified these terms.

2.9 The authors mention "weighted backprojection" in their discussion of image processing. Does RELION actually employ this procedure?

Yes. Marginalization is a form of weighted backprojection, and this is now stated explicitly.

2.10 Typos: "explaining previous observation", "aerobic glycolysis", "phosphorylation of E1 by kinases and phosphatases", "withithe"

Thanks; fixed.

Reviewed by Serban Ilca

Reviewer #3:

The authors have made an extensive effort to characterize the inner core structure of *N. crassa* pyruvate dehydrogenase complex (PDC) that is formed by the C-terminal oligomer-forming, domains of the E2 and protein X (PX) components (here E2i and PXi). E2i forms a typical 60 domain icosahedral dodecahedron structure with 20 trimers at the corners and 12 pentagonal faces via 2-fold interactions among trimers. Like mammalian PDC, *N. crassa* PX is shown to bind E3 and this retention of E3 is required for PDC function. Similar to *S. cerevisiae* PDC inner core, the PXi domain is shown to be held inside the E2i dodecahedron; this is in marked contrast to mammalian (human best characterized) PDC inner core in which E3BP inner domains substitute for the related E2i domains within the dodecahedron. The authors effectively show that PXi form trimers and binding of 4 PXi trimers within the dodecahedron employs a different symmetry (tetrahedral rather than icosahedral) and that this remains

Erik Lindahl
Professor of Biophysics

the case when linker region held outer domains (lipoyl. and subunit binding) are removed by truncation. Likely requirements for gaining and retaining PXi domains bound to E2i oligomer are discussed (more below). The studies are thorough, well performed and a substantial advance.

Most of the following review analysis involves requesting clarifications and deserve responses.

3.1 While the essential role of PX for retention of E3 with the complex is established, it would seem that a requirement of PX for some PDC function is not since studies were not conducted with excess E3 added to complexes lacking PX (PX knockout preparations).

*We agree that PX has not been conclusively shown to be essential for retained PDC function or sucrose metabolism. Free or weakly bound E3 (through e.g. the E2 PSBD) might be sufficient for retained but impaired pyruvate metabolism through the PDC in vivo (supported by our results, shown in figure S9). We have clarified this by stating that **“Future studies are however necessary to investigate to what extent free E3 can rescue function.”***

3.2 A tetrahedral arrangement involving 4 trimers of PXi with equivalent physical positioning is indicated. This is implicitly indicated but never explicitly stated which would aid a nonexpert readers. Besides physical constraints on space and the nature of interactions of PXi trimer subunits with E2i at 2-fold axis within its icosahedral structure, the presence, absence, or unknown status of any interaction between trimers needs to be explicitly addressed. The likelihood for such interactions could be exhibited more clearly with removal of E2i dodecahedron (leaving 4 trimers) along with insertion of a tetrahedral diagram (assume corners of which would be at central holes of trimers). Only part of the PXi domain structure was resolved; the possibility that nonresolved parts of the domain trimers could participate in inter trimer interactions that help maintain the tetramer arrangement should be part of this evaluation.

*Such a representation would indeed be helpful, and has now been included as panel e) of Fig. S1, which is referenced in the text. An explicit mention of four-threefold oligomers is also made in the caption of Fig. 1. We observe no direct interaction between PX trimers, and expect that they do not interact or stabilize each other. As the reviewer suggests, we therefore now include an explicit postulate of absence of trimer-trimer interactions: **“In neither of these classes do the interior trimeric assemblies appear to come into physical contact.” “We find that no two E2-trimers separated by fewer than two intermediate-E2 trimers within the core scaffold can accommodate a structured interior basket density (Fig. S1D).” “As such, we find that interior assemblies do not need to make contact to arrange as observed, and therefore postulate that they do not.”***

3.3 Indications of motion of bound trimers are not indicated even though the E2i-binding of BH segment of Pxi at 2-fold axes appear mobile as considered in the Discussion. The degree to which the involvement of 3BH binding per trimer could limit this motion should be considered and, if likely, role inter-trimer interactions. Breathing (expansion/contraction) was found in studies of other inner core E2 oligomers with the primary change along 2-fold axes. For comparison, a critical experimental step is not described in this work. What temperature was the grid and E2 or E2-PX complexes at prior to being rapidly frozen? Expansion/contraction of inner cores is only detected with the T elevated (e.g. RT) and not with low temperature solutions of components (e.g. ice-maintained temperatures).

*Environment conditions during vitrification have been added to the methods section. We do not expect that core “breathing” influences PX arrangement, although temperature could of course influence PX flexibility. We agree that such an investigation of PX flexibility modes could be an interesting future study, but assess that without confident a) assignment of the BH to the primary sequence and b) fold of the OD, any such analysis is highly speculative. This is now explicitly mentioned in the text: **“Its small size, the high noise contributed by the flexible periphery, and the difficulties associated with***

Stockholm
University

Erik Lindahl
Professor of Biophysics

symmetry unfortunately preclude a complete analysis of its flexibility and this restraint using our data and reconstruction.”

3.4 Similarly, the temperature at which PDC activity assays are conducted is not described. Is it possible that the low activity of complexes is due to exposure in cells/mitochondria to pyruvate (derived from sucrose) along with TPP but in the absence of non-acetylated CoA which leads to inactivation of E1 with most sources of complex. This is why adding enzyme or pyruvate last is important in performing activity assays.

Activity assays were conducted at room temperature, which is now properly described in the methods section (3.5). We are not aware of a mechanism by which inhibition of E1 would proceed in the presence of Pyruvate and TPP but absence of CoA, and our experiments show that addition of any substrate as the final reactant produces similar activity. It is more likely that any such discrepancy arises from E1 inactivation by phosphorylation of the endogenous preparation, which is also why we include a statement saying that phosphatase or dichloroacetate was not used to activate E1.

Minor:

3.5 In Figure 3b, the binding domain for E2 should be labeled E1BD not E3BD.

The figure has been changed to label the domain “PSBD”, which is used in the literature as the generic descriptor to contrast an E3-specific binding domain.

3.6 The acidic/hydrophobic M3 is flanked by classic linker region sequences but closest in by runs of lysine residues. Assuming this segment does play the BH role as suggested, is any role proposed for these residues and what would prevent these basic residues from disrupting interactions of M3 with basic residues in 2-fold E2i domain bridges?

*The limited structural information of PX binding limits our interpretation of the highly interesting MSAs and PX sequence. For instance, *N. crassa* does not obey the lysine-rich consensus motif following M3 (Fig S7), making any interpretation of this in terms of our reconstruction even more difficult. We have alluded to this in the text by adding the statement “**Multiple additional conservation patterns however also exist, but these unfortunately require a better structural assignment of the PX primary sequence to be confidently interpreted.**”*

3.7 Indicate reference (pdb source) for Fig. 2d. Reference 18 does indicate a distinct protein (called protein X), but this source provides no evidence for E3 binding role. A reference to this discovery is needed.

Good catch. The accession code indicated in panel d) is now also present in the caption. A reference to Powers-Greenwood (1988) and Rahmatullah (1989) has been added, which first reported E3-binding of PX (later renamed E3BD in mammals).

3.8 Despite the mammoth amount of sequence data conveyed in Figure S7, it is not clear to this nonexpert, the degree to which the findings extend to other fungi. There appear to be 13 classes of Pezizomycotina (wiki). Are the three (besides Sordariomycetes which includes Neurospora) broadly representative of classes or particularly close to Sordariomycetes? Clearly Saccharomycetes are substantially different as are the undefined “Various” fungi (include Taphrinomycotina?).

The presented subdivision of pezizomycotina into SOR/LEO/EUR/DOT/SAC reflects an existing and widely used but unranked taxonomic level reflecting natural groups within pezizomycotina but not strict taxonomic classes. This is e.g. the subdivision of Ascomycota in the PFAM entry

Erik Lindahl
Professor of Biophysics

*“E3_binding”(PF02817) from which we began our phylogenetic analysis. Our use of “class” stems from PFAM, but was formally incorrect, and has been changed to “groups”. Taphrinomycotina does not appear to be annotated to have an E3-binding protein according to PFAM, which probably indicates a limitation of the PFAM annotation, as Taphrinomycotina is extremely likely to recruit peripheral PDC enzymes E1 and E3 through on or more PSBDs. Our phylogenetic analysis is thus limited, but also only intended to convey overall conservation patterns and associated correlatives within ascomycota. We have added a remark to this effect in the methods section covering the bioinformatics: “**All retained sequences had been taxonomically annotated by PFAM as one of the 5 Ascomycota “classes” Sordariomycetes (Sor), Eurotiomycetes (Eur), Leotiomycetes (Leo), Dotiideomycetes (Dot) or Saccharomycetes (Sac). These are not true taxonomic classes, and are thus referred to as “groups” in the present work (Fig. S7). [...] We were unable to find any CBD-homologous sequences outside of Ascomycota.**”*

Comments:

3.9 This reviewer agrees with the Discussion comments indicating that assembly of oligomers of PXi domains occurs within the dodecahedron since trimers could not fit through the five-fold faces and that assembly is probably required for retaining PXi domains in this space

3.10 While agreeing that sequences of *S. cerevisiae* and *N. crassa* PXi are at best distantly related (Fig. 7S), the authors are generous regarding the putative findings of prior studies with *S. cerevisiae* PXi binding in larger amounts to E2i oligomer. Cryo-EM results were only analyzed using icosahedral symmetry (possibly giving misleading results like Fig. 1a) and the 1996 paper (ref. 24) suggesting higher stoichiometric binding of truncated PXi does not provide extant supporting data with the many needed controls, but it only supplies a minimal Table that must be accepted based on faith. These limitations could be indicated.

*The reviewer correctly points out that previous findings did not have the benefit of the present structural view to rationalize results. Barring evidence to the contrary we however believe we should rely on their presented data, and offer a reconciling model, even if it's a bit generous. We agree with the reviewer that this should be more directly addressed in the text, and have thus amended the discussion by stating that “**While our reconstruction invites a direct interpretation of previous findings, further biochemical investigation of binding stoichiometry that consider it explicitly will be necessary to corroborate the structural regulation of PDC composition suggested. The discrepancy in sequence identity and conservation comparing the pezizomycotina and saccharomycotina CBD (Fig. S7) also merit future research to completely understand previous findings. The presented mode of PX-binding in N. crassa should greatly facilitate this.**”*

Once more, thanks a lot for three thorough reviews that all helped improve the study!

Sincerely,

Erik Lindahl

REVIEWERS' COMMENTS:

Reviewer #1 (Remarks to the Author):

This revised manuscript provides new and useful information about a key metabolic complex, pyruvate dehydrogenase (PDC) in a not well studied fungal family. The analyses provide structural examples through cryoEM for the fungal version. Some features found are similar to those seen in related complexes from various species, while others are new and somewhat unexpected. Together, they expand our knowledge of general PDC function and environmental adaptability. As a revised submission, the authors generally did a good job at addressing previous comments in the earlier critique. Textual modifications were made, and in some cases new figures were added as was suggested. The authors took the critiques to heart, leading to an improved manuscript. I have just one comment regarding the earlier critique and response.

in response to the comment

"Finer control over metabolic flux is also afforded by adjusting the relative proportions of the PDC components, regulating its overall activity" is ambiguous. Is this suggesting a DYNAMIC change in composition during the proteins functioning, or a static change that differs in species (or tissue) and is maintained throughout? The authors have now included a statement ""The saturation and relative proportions of tethered E1 and E3 may also be subject to change in response to external cues and alteration in metabolic requirements."

It's still not clear regarding the temporal nature of the change, I.e are the "external cues" acted on during initial complex formation leaving the once formed complex intact thereafter, or does it mean the complex actually dissociates and reforms changing components repeatedly, once the individual components have been expressed?

Reviewer #2 (Remarks to the Author):

The authors did a fantastic job addressing the points I raised as well as updating their manuscript, and I would like to congratulate them on this. In my opinion, the manuscript is now significantly easier to understand and follow. I do not have any outstanding major comments or suggestions regarding any aspect of the paper.

Serban Ilca

Reviewer #3 (Remarks to the Author):

Reviewer 3

The authors have made a diligent and thorough response to review comments. This reviewer is satisfied with the responses to my prior review (with minor exception below) and I support publication of this work.

The following response is to supply useful information to the authors and while it contains a question does not require a response.

Authors Response 3.4

We are not aware of a mechanism by which inhibition of E1 would proceed in the presence of Pyruvate and TPP but absence of CoA, and our experiments show that addition of any substrate as the final reactant produces similar activity. It is more likely that any such discrepancy arises from E1 inactivation by phosphorylation of the endogenous preparation, which is also why we include a

statement saying that phosphatase or dichloroacetate was not used to activate E1.

Reviewer

The earliest evidence for marked reduction in complex activity due to altered E1 following incubation with low pyruvate and TPP came from the research of L.S. Khailova and colleagues (couple references below). The initial modification involved acetylation of a cysteine. Later Cys62 was identified in sequence of mature mammalian E1 α as being functionally important (aligns with an identical neighboring sequence IRGFCHL in *Neurospora crassa* E1 α). However, though this Cys in human E1 α is positioned next to function part of the thiazole ring of TPP, I am not aware of studies showing this Cys is the one acetylated. Though loss of activity was initially partially reversible with thiol treatment, the acetyl group tended to move to an amine. Once activity is reduced over a significant period of time, it does not matter about the order of addition of substrates in assays. Phosphorylation-dephosphorylation was definitely not involved; an area this reviewer is knowledgeable about (>40 publication on PDK-PDP function). This reviewer is more familiar with assays at 30 deg. C for mammalian enzymes for which assays of overall activity with limiting E1 should give specific activities >24 $\mu\text{mole}/\text{min}\cdot\text{mg}$ E1 and, with 20 plus E1 tetramers per complex, specific activities of per mg complex >13 $\mu\text{mole}/\text{min}$. However, it has proved very difficult using recombinant expression of E1 in *E. coli* to reach specific activities this high (50% is good). Did the MS analysis used to detect *N. crassa* E1 α subunit (Results: 1.2) have high enough resolution to detect any acetylation?

The substrate-mediated inactivation of the pyruvate dehydrogenase component of the pigeon breast muscle pyruvate dehydrogenase complex. Khailova LS, Nemerya NS, Severin SE. *Biochem. Int.* 1983 Oct;7(4):423-32.

Substrate-dependent inactivation of muscle pyruvate dehydrogenase: identification of the acetyl-substituted enzyme form. Khailova LS, Alexandrovitch OV, Severin SE. *Biochem. Int.* 1985 Feb;10(2):291-300.

Erik Lindahl
Professor of Biophysics

Re: NCOMMS-20-14293A

Dear Katarzyna,

Thank you so much for the positive response, acceptance of the MS, and not least a *very* positive & constructive review process. The remaining two comments from reviewers 1 & 3 were good points that we have addressed in the new version (our responses below in green). We have also complied with the editorial requests and attached the requested checklists.

Reviewer #1 (Remarks to the Author)

This revised manuscript provides new and useful information about a key metabolic complex, pyruvate dehydrogenase (PDC) in a not well studied fungal family. The analyses provide structural examples through cryoEM for the fungal version. Some features found are similar to those seen in related complexes from various species, while others are new and somewhat unexpected. Together, they expand our knowledge of general PDC function and environmental adaptability. As a revised submission, the authors generally did a good job at addressing previous comments in the earlier critique. Textual modifications were made, and in some cases new figures were added as was suggested. The authors took the critiques to heart, leading to an improved manuscript. I have just one comment regarding the earlier critique and response.

in response to the comment

“Finer control over metabolic flux is also afforded by adjusting the relative proportions of the PDC components, regulating its overall activity” is ambiguous. Is this suggesting a DYNAMIC change in composition during the proteins functioning, or a static change that differs in species (or tissue) and is maintained throughout? The authors have now included a statement “The saturation and relative proportions of tethered E1 and E3 may also be subject to change in response to external cues and alteration in metabolic requirements.” It’s still not clear regarding the temporal nature of the change, I.e are the “external cues” acted on during initial complex formation leaving the once formed complex intact thereafter, or does it mean the complex actually dissociates and reforms changing components repeatedly, once the individual components have been expressed?

We meant to convey that the proportion of bound E1 and E3 may be dependent on reorganization or expression level, and that the composition of the PDC in this sense is not fixed, or expected to be strict. We should have made this clearer, and so now state that **“The saturation and relative proportions of tethered E1 and E3 may also be influenced by availability of tethered or tethering components, which in turn may change in response to expression levels and and alteration in metabolic requirements.”**. This omits the formulation “external cues”, which we agree was ambiguous.

Reviewer #3 (Remarks to the Author):

The authors have made a diligent and thorough response to review comments. This reviewer is satisfied with the responses to my prior review (with minor exception below) and I support publication of this work.

Department of Biochemistry and Biophysics

Stockholm University
Science for Life Laboratory
Box 1031
SE-171 21 Solna, Sweden

Visiting address:
Science for Life Laboratory
Tomtebodavägen 23A, Solna

Phone: +46-734618050
Cell: +46-734618050
Mail: erik.lindahl@dbb.su.se

Erik Lindahl
Professor of Biophysics

The following response is to supply useful information to the authors and while it contains a question does not require a response.

Authors Response 3.4

We are not aware of a mechanism by which inhibition of E1 would proceed in the presence of Pyruvate and TPP but absence of CoA, and our experiments show that addition of any substrate as the final reactant produces similar activity. It is more likely that any such discrepancy arises from E1 inactivation by phosphorylation of the endogenous preparation, which is also why we include a statement saying that phosphatase or dichloroacetate was not used to activate E1.

Reviewer

The earliest evidence for marked reduction in complex activity due to altered E1 following incubation with low pyruvate and TPP came from the research of L.S. Khailova and colleagues (couple references below). The initial modification involved acetylation of a cysteine. Later Cys62 was identified in sequence of mature mammalian E1alpha as being functionally important (aligns with an identical neighboring sequence IRGFCHL in Neurospora crassa E1alpha). However, though this Cys in human E1 alpha is positioned next to function part of the thiazole ring of TPP, I am not aware of studies showing this Cys is the one acetylated. Though loss of activity was initially partially reversible with thiol treatment, the acetyl group tended to move to an amine. Once activity is reduced over a significant period of time, it does not matter about the order of addition of substrates in assays. Phosphorylation-dephosphorylation was definitely not involved; an area this reviewer is knowledgeable about (>40 publication on PDK-PDP function). This reviewer is more familiar with assays at 30 deg. C for mammalian enzymes for which assays of overall activity with limiting E1 should give specific activities >24 $\mu\text{mole}/\text{min}\cdot\text{mg}$ E1 and, with 20 plus E1 tetramers per complex, specific activities of per mg complex >13 $\mu\text{mole}/\text{min}$. However, it has proved very difficult using recombinant expression of E1 in E. Coli to reach specific activities this high (50% is good). Did the MS analysis used to detect N. crassa E1alpha subunit (Results: 1.2) have high enough resolution to detect any acetylation?

This is indeed a good idea! Unfortunately acetylation was not included in the original PTM analysis since we did not anticipate acetylation to any significant or relevant degree. We anticipate that the MS analysis could resolve acetylation if present, so it's something we will consider to pursue in the future, and we have also added a brief comment about this in the methods section to aid other researchers - thanks for the suggestions!

Best regards,

/Erik Lindahl/